# Beyond Surface Structure: A Causal Assessment of LLMs' Comprehension Ability

**Yujin Han**[1,2]*, **Lei Xu**[1], **Sirui Chen**[3], **Difan Zou**[2], **Chaochao Lu**[1]†

[1]Shanghai Artificial Intelligence Laboratory, [2]The University of Hong Kong, [3]Tongji University
yujinhan@connect.hku.hk, dzou@cs.hku.hk,
{xulei, chensirui, luchaochao}@pjlab.org.cn

## Abstract

Large language models (LLMs) have shown remarkable capability in natural language tasks, yet debate persists on whether they truly comprehend deep structure (i.e., core semantics) or merely rely on surface structure (e.g., presentation format). Prior studies observe that LLMs' performance declines when intervening on surface structure, arguing their success relies on surface structure recognition. However, surface structure sensitivity does not prevent deep structure comprehension. Rigorously evaluating LLMs' capability requires analyzing both, yet deep structure is often overlooked. To this end, we assess LLMs' comprehension ability using causal mediation analysis, aiming to fully discover the capability of using both deep and surface structures. Specifically, we formulate the comprehension of deep structure as direct causal effect (DCE) and that of surface structure as indirect causal effect (ICE), respectively. To address the non-estimability of original DCE and ICE — stemming from the infeasibility of isolating mutual influences of deep and surface structures, we develop the corresponding quantifiable surrogates, including approximated DCE (ADCE) and approximated ICE (AICE). We further apply the ADCE to evaluate a series of mainstream LLMs (and the one with random weights), showing that most of them exhibit deep structure comprehension ability, which grows along with the prediction accuracy. Comparing ADCE and AICE demonstrates closed-source LLMs (e.g., GPT) rely more on deep structure, while open-source LLMs (e.g., Llama) are more surface-sensitive, which decreases with model scale. Theoretically, ADCE is a bidirectional evaluation, which measures both the sufficiency and necessity of deep structure changes in causing output variations, thus offering a more comprehensive assessment than accuracy, a common evaluation in LLMs. Our work provides new insights into LLMs' deep structure comprehension and offers novel methods for LLMs evaluation. The code for our project is available at ADCE Project.

## 1 Introduction

Large language models (LLMs) have demonstrated unprecedented capability in various natural language tasks (Achiam et al., 2023; Touvron et al., 2023a;b; Chowdhery et al., 2023; Anil et al., 2023; Team et al., 2023). Despite these achievements, there remains a debate over whether LLMs truly grasp the deep structure necessary for solving variations of the same problem, or if they simply learn the surface structure present in data. The distinction between surface and deep structure, defined in surface structure theory (Chomsky et al., 1971), differentiates between observable sentence forms and the underlying semantic units that represent a question's core meaning. This distinction is further illustrated with examples in Table 1. Many studies evaluating LLMs based on task-specific accuracy (Zeng et al., 2023; Wang et al., 2023; Chan et al., 2023) often neglect their capacity to understand deep structures leading to correct solutions. This oversight may mislead model performance, as high accuracy might stem from learning surface structures in training data instead of

---

*This work was done when Yujin was an intern at Shanghai Artificial Intelligence Laboratory.
†Corresponding author.

Table 1: Examples of two-digit multiplication with interventions on deep and surface structures: deep structure embodies core semantics (e.g., numbers and operators), while surface structure encompasses linguistic forms (e.g., question format). Among given intervention strategies, changes in deep structure inherently alter surface structure. More examples on both structures in Appendix A.

| Example Questions | Deep & Surface Intervention | Surface Intervention Only | Strategy |
|---|---|---|---|
| What is 50 times 20 ? A:1000 | `What is ⟨Mask⟩ times 20?` `A:None` | `What is 50 times 20⟨Mask⟩` `A:1000` | *Mask* |
| | `How much is 10 multiplied` `by 50?  A:500` | `How much is 20 multiplied` `by 50?  A:1000` | *Rephrase* |
| | `What is * times 20?` `A:None` | `What * 50 times 20?` `A:1000` | *Replace* |
| | `50 is What times 20?` `A:2.5` | `is What 50 times 20?` `A:1000` | *Swap* |

deep structure. Such learning can lead spurious correlations between inputs and responses, limiting generalization to novel and realistic scenarios (Guo et al., 2024; Jiang et al., 2024b).

Recent studies tend to understand surface structure beyond accuracy and indicate LLMs predominantly rely on surface structure to generate responses (Stolfo et al., 2022; Hooda et al., 2024; González & Nori, 2024; Guo et al., 2024; Jiang et al., 2024b). Interventions unrelated to answers, like renaming entities (Jiang et al., 2024b) or swapping code blocks (Hooda et al., 2024), decrease performance. This sensitivity to minor input changes suggests LLMs' task performance depends more on surface structure recognition (Hooda et al., 2024; Jiang et al., 2024b).

However, prior work has primarily focused on LLMs' sensitivity to surface structure, without adequately examining their comprehension of deep structure. While sensitivity to surface-level interventions shows a lack of robustness to superficial changes, it does not necessarily preclude an understanding of deep structure. To ascertain whether LLMs are merely surface structure learners, a comparative analysis of their understanding of both deep and surface structures is essential, which has been largely overlooked in current research. To validate this hypothesis, we conduct the following experiment. Initially, LLMs reason on the complete dataset to identify correctly answered samples. Subsequently, using *Mask* strategy (Table 1), we create two intervention groups from the identified correct samples: one with interventions to both deep and surface structures, and another with only surface interventions. We then evaluate these intervened samples and compare the accuracy declines (Figure 1). We observe that surface-only interventions cause slight accuracy decline, while

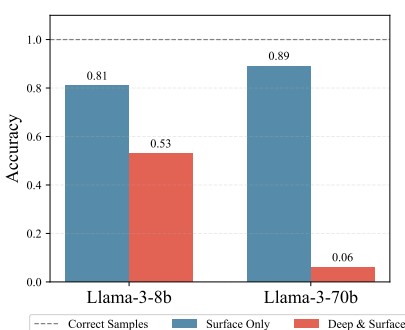

Figure 1: Surface structure interventions cause subtle accuracy degradation relative to the obvious accuracy decline from deep structure changes.

combined surface and deep modifications result in significant performance degradation. This challenges the prevailing assumption that LLM responses are predominantly based on surface structure and suggests a more significant reliance on deep structure. Given above observation and the prevalent oversight of deep structure understanding, we propose a fundamental research question:

*Do LLMs genuinely comprehend deep structure for problem-solving, or do they primarily rely on learning surface structure?*

To address the issue, corresponding metrics are required, which should: (1) Quantify LLMs' understanding capabilities of deep and surface structures; (2) Be widely applicable across diverse tasks and LLMs, overcoming limitations of previous methods restricted to specific tasks (e.g., data flow problems in programming (Hooda et al., 2024), divisibility issues in mathematics (González & Nori, 2024)), specific data types (e.g., synthetic data with fixed textual templates (Jiang et al., 2024b)), or specific models (e.g., small-sized transformers trained from scratch (Jin & Rinard)).

In this paper, we employ causal mediation analysis (Imai et al., 2010a;b; Hicks & Tingley, 2011) to formulate LLMs' deep structure comprehension as the direct causal effect (DCE) of deep structure on outputs, and surface structure comprehension as the indirect causal effect (ICE) of surface structure on outputs. However, estimating DCE and ICE requires isolating the mutual influences

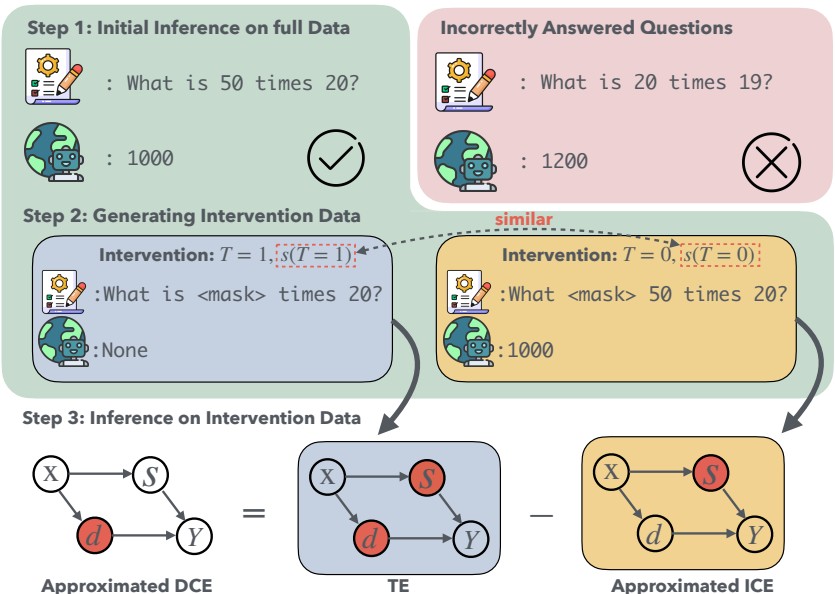

Figure 2: Approximated DCE (ADCE) quantifies LLMs' deep structure comprehension, while approximated ICE (AICE) measures surface structure understanding. Comparing them reveals LLMs' reliance on deep or surface structures. Our method involves: initial inference, intervention on correct samples, and secondary inference for ADCE and AICE calculation. More details are in Appendix D.

between deep and surface structures, which is infeasible, e.g., the impossibility of modifying deep structure without altering surface structure. Consequently, we propose approximated DCE (ADCE) and approximated ICE (AICE) as proxies for DCE and ICE. ADCE and AICE empirically quantify LLMs' deep and surface structure comprehension across diverse tasks, revealing that LLMs' understanding beyond surface structures. Our method is widely applicable, independent of data or model constraints, thus suitable for diverse tasks and models. We summarize our key contributions as:

*Methodologically*, we formalize LLMs' deep structure comprehension ability based on causal mediation analysis and propose an estimable approximated direct causal effect (ADCE) to quantify this ability. The proposed method also includes the approximated indirect causal effect (AICE) of surface structure, enabling comparison of LLMs' reliance on deep and surface structures (in Section 3).

*Empirically*, we evaluate deep structure comprehension in mainstream LLMs across tasks, revealing widespread deep understanding that strongly correlates with accuracy (in Section 4.2). Further comparison between ADCE and AICE shows tested closed-source LLMs excel in deep comprehension, while tested open-source LLMs shift from surface to deep understanding with scale (in Section 4.4).

*Theoretically*, we prove ADCE evaluates both sufficiency and necessity of deep structure changes in output variations (in Section 3.4), which offers a bidirectional assessment of LLM performance beyond output correctness, in contrast to the simple criteria like prediction accuracy. This theoretical point is supported by subsequent spurious correlation experiments (in Section 4.5). This suggests that ADCE can serve as a more comprehensive assessment criterion to evaluate and understand the ability of LLMs (e.g., the dependence of LLM outputs on the core semantics of the inputs).

## 2 A CAUSAL PERSPECTIVE OF LLMS' COMPREHENSION ABILITY

In this section, we define LLMs' deep structure comprehension ability by formulating it as a problem of estimating causal effects. We first introduce important notations for subsequent analysis. Consider a dataset $\mathcal{D} = \{(\boldsymbol{x}_i, y_i)\}_{i=1}^{n}$, where $\boldsymbol{x}_i$ denotes the $i$-th question and $y_i$ represents the corresponding answer. Each question $\boldsymbol{x}_i := (d_i, s_i)$ can be split into two independent components (Stolfo et al., 2022): the deep structure $d_i$ and the surface structure $s_i$, with $d_i \perp\!\!\!\perp s_i | \boldsymbol{x}_i$. Given an LLM parameterized by $\theta \in \Theta$, denoted as $f_{\boldsymbol{\theta}}$, its output for $\boldsymbol{x}_i$ is represented as $Y_i(\boldsymbol{x}_i) := f_{\boldsymbol{\theta}}(\boldsymbol{x}_i)$.

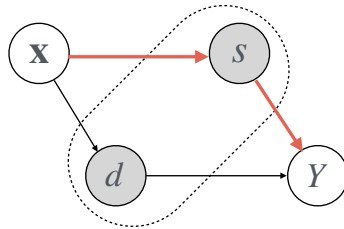

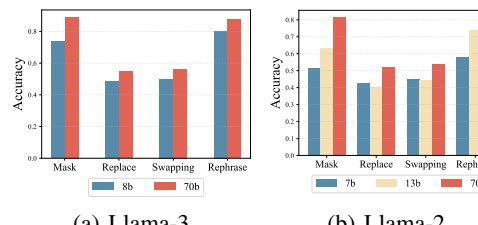

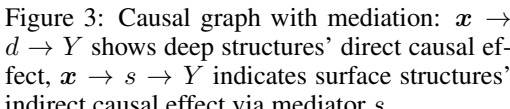

(a) Llama-3       (b) Llama-2

Figure 3: Causal graph with mediation: $\boldsymbol{x} \to d \to Y$ shows deep structures' direct causal effect, $\boldsymbol{x} \to s \to Y$ indicates surface structures' indirect causal effect via mediator $s$.

Figure 4: For the four intervention strategies, LLM accuracy drops from 100% when surface structures are altered while deep structures remain unchanged in initially correct samples.

**Comprehension Ability.** While high accuracy often indicates a high-performing model, our work delves into whether LLMs achieve this accuracy through a genuine understanding of deep structure. We propose that an LLM, $f_{\boldsymbol{\theta}}$, acting as a "deep thinker", should not only provide correct answers but also fundamentally depend on deep structure for responses. Formally, let $\mathcal{D}_c \subseteq \mathcal{D}$ be a subset of questions correctly answered by $f_{\boldsymbol{\theta}}$. An LLM $f_{\boldsymbol{\theta}}$ possesses deep structure comprehension satisfy

$$\mathbb{1}_{Y(\boldsymbol{x}'_i)=y_i} = \begin{cases} 0, & \forall d'_i \neq d_i \\ 1, & \forall d'_i = d_i \end{cases} \tag{1}$$

where $\mathbb{1}$ means the indicator function, the modified $\boldsymbol{x}'_i = (d'_i, s'_i)$ and the original $\boldsymbol{x}_i = (d_i, s_i)$. Note that, the surface structures $s_i$ and $s'_i$ may be identical or different. In other words, the output of the model $f_{\boldsymbol{\theta}}$ should only be altered by changes in the deep structure $d_i$, underscoring the model's reliance on deep rather than surface structure for generating responses.

Equation 1 quantifies an LLM's comprehension of deep structure by comparing outputs following changes to corresponding structures. This inspires a causal effect estimation perspective, where changes in outputs are viewed as different potential outcomes (Pearl, 2001; Rubin, 2005), resulting from interventions on either deep or surface structures.

**Causal Effect Estimation.** We proceed by defining LLMs' comprehension ability as a causal effect estimation problem. Define the treatment assignment variable $T$ on input $\boldsymbol{x}_i$ as:

$$T = \begin{cases} 0 & \text{intervention alters } s_i, \text{ preserves } d_i \\ 1 & \text{intervention alters both } s_i \text{ and } d_i \end{cases} \tag{2}$$

Both $d_i$ and $s_i$ are unobservable, non-manipulable latent variables. Intervention $T$ only manipulate the observable input $\boldsymbol{x}_i$. The potential outcome for $\boldsymbol{x}_i$ under $T = t$ is $Y_i(t)$. The deep structure comprehension ability is defined as the causal effect of deep structure on an LLM's output, i.e., the expected change in the output when intervening on the deep structure while keeping surface structure fixed. Analogously, the surface structure comprehension capability is defined.

By defining LLMs' deep and surface structure comprehension as causal effects, we establish a causal estimation framework. Leveraging this framework, we quantify abstract comprehension capabilities via estimable causal effects, enabling objective assessment of LLMs' understanding.

## 3 METHOD

This section focuses on the causal effect of deep structure on output, as defined in Section 2. Notably, estimating this causal effect inherently requires quantifying the causal effect of surface structure. Thus, by concentrating on deep structure, we also gain insights into the surface structure. Section 3.1 presents a causal graph linking inputs, structures, and outcomes, formulating comprehension as direct (DCE) and indirect causal effects (ICE). Section 3.2 further addresses the non-estimability of DCE and ICE by proposing their approximations: ADCE and AICE. To estimate these metric in practice, Section 3.3 details the generation of intervention data necessary for estimating ADCE and AICE. Finally, to demonstrate the value of our metric in LLMs evaluation, Section 3.4 shows how ADCE outperforms the common metric, accuracy, in evaluating LLMs' deep structure dependency.

### 3.1 FORMULATING DEEP STRUCTURE COMPREHENSION AS DIRECT CAUSAL EFFECT

Figure 3 presents a causal graph with mediation depicting relationships among inputs $\boldsymbol{x}$, deep structure $d$, surface structure $s$, and outcome $Y$. It illustrates how $\boldsymbol{x}$ influences $Y$ via $d$ ($\boldsymbol{x} \to d \to Y$)

Table 2: Examples of different intervention strategies on mathematics and common sense tasks. More illustrations on multiple tasks are included in Appendix F.1.

| Dataset | Term | Origin & Intervention Data |
|---|---|---|
| 2-digit Multiplication (Mask) | Origin | `What is 50 times 20?  A: 1000` |
| | TE with $T = 1, s(T = 1)$ | `What is <Mask> times 20?  A: None` |
| | AICE with $T = 0, s(T = 0)$ | `What <Mask> 50 times 20?  A: 1000` |
| CommonsenseQA (Rephrase) | Origin | `Reading newspaper one of many ways to practice your what?  A: literacy` |
| | TE with $T = 1, s(T = 1)$ | `Using newspapers to wrap gifts is one way to practice your what?  A: money` |
| | AICE with $T = 0, s(T = 0)$ | `Using newspapers to read articles is one way to practice your what?  A: literacy` |

and $s$ ($\boldsymbol{x} \to s \to Y$).Deep structure, reflecting core semantics, logically correlates with output, justifying the path $\boldsymbol{x} \to d \to Y$. Surface structure's impact on output is considered for the following reasons: Existing studies show surface structure changes affect LLMs outcomes even with constant deep structure (Stolfo et al., 2022; Hooda et al., 2024; Jiang et al., 2024b; Guo et al., 2024). Our two-digit multiplication experiment in Figure 4 confirms this, showing performance decline on corrected answered samples when modifying only surface structure.

Figure 3 illustrates a causal mediation analysis, focusing on the direct causal effect (DCE) of deep structure $d$ on output $Y$ via the path $\boldsymbol{x} \to d \to Y$. The required assumptions for causal mediation analysis — *positivity*, *consistency*, and *sequential ignorability* (Rubin, 1974; VanderWeele & Vansteelandt, 2009; Cole & Frangakis, 2009; Coffman et al., 2021; Nguyen et al., 2022) — are satisfied, as detailed in Appendix B.1. This analytical setup allows us to rigorously examine the influence of deep structure on model outputs, isolating it from the effects of surface structure.

As directly estimating DCE is intractable due to challenges in altering deep structure while maintaining surface structure, an indirect method has been developed (Pearl, 2001; Imai et al., 2010a;b; VanderWeele, 2013; Richiardi et al., 2013), estimating DCE as:

$$\underbrace{\delta_{\text{DCE}}}_{\text{DCE}} = \underbrace{\mathbb{E}_{\boldsymbol{x}_i}[Y_i(T = 1, s(T = 1)) - Y_i^{\text{origin}}]}_{\text{TE}} - \underbrace{\mathbb{E}_{\boldsymbol{x}_i}[Y_i(T = 0, s(T = 1)) - Y_i^{\text{origin}}]}_{\text{ICE}} \quad (3)$$

where $s(T = t)$ is the mediator value at $T = t$. For $\mathbf{x}_i$, $Y_i(T = 1, s(T = 1))$, $Y_i(T = 0, s(T = 1))$, and $Y_i^{\text{origin}}$ represent outcomes with both structures altered, only surface changed, and unintervened original structures, respectively. Equation 3 specifically emphasizes the effect of deep structure on the output while maintaining the surface structure constant at $s(T = 1)$. ICE in Equation 3 via $\mathbf{x} \to s \to Y$ quantifies the causal effect of surface structure on $Y$. ICE and DCE comprise the total effect (TE) of $\mathbf{x}$ on $Y$. Appendix B.2 provide more details on DCE, ICE, and TE.

### 3.2 ESTIMATING DCE FROM DATA: CHALLENGES AND SOLUTIONS

Although Equation 3 can indirectly esitimate DCE, it still suffers the following issues:

- Unobservability: ICE in Equation 3 is unobservable due to a paradox: The surface structure in ICE must maintain the value it would have under deep structure change ($s(T = 1)$), while the deep structure in ICE should remain unchanged ($T = 0$). Consider 2-digit multiplication task in Table 1, ICE should preserve the surface query format as *What is <mask> times 20?* ($s(T = 1)$) where the deep structure is altered ($T = 1$), thereby contravening the condition $T = 0$.
- Incomputability: Equation 3 requires differencing $Y_i$ and $Y_i^{\text{origin}}$, but the outputs of LLMs typically lack numerical form, complicating the execution of such subtraction. For instance, in word unscrambling tasks (bench authors, 2023), the string nature of outputs inherently prevents direct arithmetic operations such as subtraction.

To address above issues in DCE, we propose the following solutions. Based on these solutions, we derive the approximated direct causal effect (ADCE) as an estimable surrogate for DCE.

**Addressing Unobservability.** ICE in Equation 3 requires simultaneous $T = 0$ and $s(T = 1)$, which are unobservable in practice. Therefore, we propose approximated DCE (ADCE) to substitute origi-

nal ICE in Equation 3 with observable $(T = 0, s(T = 0))$ as approximated ICE (AICE). The efficacy of this approximation hinges on the similarity between the original ICE and AICE, specifically the similarity between $(T = 0, s(T = 1))$ and $(T = 0, s(T = 0))$. To ensure approximation validity, we meticulously design intervention strategies for generating data that minimize the discrepancy between the original ICE and AICE. Detailed intervention strategies are discussed in Section 3.3. The AICE and corresponding approximated DCE (ADCE) can be represented as:

$$\underbrace{\delta_{\text{ADCE}}}_{\text{approximated DCE (ADCE)}} = \underbrace{\mathbb{E}_{\boldsymbol{x}_i}[Y_i(T = 1, s(T = 1)) - Y_i^{\text{origin}}]}_{\text{TE}} - \underbrace{\mathbb{E}_{\boldsymbol{x}_i}[Y_i(T = 0, s(T = 0)) - Y_i^{\text{origin}}]}_{\text{approximated ICE (AICE)}}$$
(4)

Observable AICE in Equation 4 quantifies surface structure's causal effect, i.e., LLMs' surface structure comprehension ability while controlling deep structure. Strategies in Section 3.3, like minimally modifying TE with $(T = 1, s(T = 1))$ to AICE with $(T = 0, s(T = 0))$, ensure Equation 4 maximizes surface similarity between TE and AICE, isolating deep structure impacts in ADCE.

**Addressing Incomputability:** To address incomputability, following (Stolfo et al., 2022; Chen et al., 2024), we introduce indicator function $\mathbb{1}$ instead of numerical differencing. Indicator function operations can capture output changes relative to the original output, making ADCE estimation applicable across diverse model outputs. We then redefine

$$\underbrace{\hat{\delta}_{\text{ADCE}}}_{\text{approximated DCE (ADCE)}} = \underbrace{\mathbb{E}_{\boldsymbol{x}_i}\left[\mathbb{1}_{Y_i(T=1,s(T=1)) \neq Y_i^{\text{origin}}}\right]}_{\text{TE}} - \underbrace{\mathbb{E}_{\boldsymbol{x}_i}\left[\mathbb{1}_{Y_i(T=0,s(T=0)) \neq Y_i^{\text{origin}}}\right]}_{\text{approximated ICE (AICE)}}$$
(5)

Moreover, as detailed in Section 2, LLMs solely utilizing deep structure for answering satisfy:

$$Y_i(T = 1, s(T = 1)) \neq Y_i^{\text{origin}} \quad \text{and} \quad Y_i(T = 0, s(T = 0)) = Y_i^{\text{origin}}.$$
(6)

Combining Equation 5 and Equation 6 yields $\hat{\delta}_{\text{ADCE}} \in [-1, 1]$, where larger values indicate stronger causal effects of deep structure on model output. It means higher $\hat{\delta}_{\text{ADCE}}$ suggests greater dependence of LLMs' outputs on deep structure, implying enhanced deep structure comprehension. Thus, $\hat{\delta}_{\text{ADCE}}$ is interpretable and enables comparisons across both tasks and models.

## 3.3 GENERATING INTERVENTION DATA FOR APPROXIMATED DCE ESTIMATION

To indirectly estimate ADCE, we should detail the generation of intervention data required for TE and AICE estimation in Equation 5. Specifically, we focus on constructing appropriate approximation to minimize the discrepancy between AICE in Equation 5 and oracle ICE in Equation 3.

**Intervention Data for TE.** TE requires intervention data with altered deep structure $(T = 1)$ and matched surface structure $(s(T = 1))$. To achieve this, we intervene on inputs $\boldsymbol{x}$ to alter core semantics using *Mask* and *Rephrase* strategies in Table 1. For inputs with explicit core semantic words, such as numbers and operators in two-digit multiplication tasks, we apply *Mask*; otherwise, we use *Rephrase* . Table 2 shows examples with diverse intervention strategies for TE.

**Intervention Data for AICE.** To approximate the unobservable ICE in Equation 3, we minimally modify the deep structure of TE with $(T = 1, s(T = 1))$ in Equation 5 to derive AICE with $(T = 0, s(T = 0))$. Deriving AICE from TE yields an observable substitute for the original ICE and ensures high similarity between $s(T = 1)$ in TE and $s(T = 0)$ in AICE. Thus, the key distinction between TE and AICE lies in the deep structure difference, ensuring isolation of surface structure's effect on output. Specially, we employ two strategies: (1) *Mask*: masking $k$ non-core semantic words closest to the masked core semantic word in TE; (2) *Rephrase*: minimizing word-level modifications to transform TE with $(T = 1, s(T = 1))$ to AICE with $(T = 0, s(T = 0))$ with prompts suck as *modify the keywords with minimal word changes* . Table 2 provides detailed intervention examples.

For rephrasing, we use Claude-3.5-Sonnet (Anthropic, 2024) and design a self-checking mechanism. Claude re-answers rephrased questions to verify deep structure alteration and preservation. Algorithm 2 outlines the process, with detailed mask rules and rephrase prompts in Appendix F.1.

## 3.4 ADCE: BIDIRECTIONAL EVALUATION OF DEEP STRUCTURE COMPREHENSION

This section compares the proposed ADCE in equation 5 with accuracy metrics. Our analysis demonstrates that ADCE better reflects the bidirectional relationship between deep structure and

model outputs, regardless of whether the outputs are depended on the deep structure or merely associated with surface structure due to spurious correlations.

**LLMs' Output Depends on Deep Structure.** When outputs of LLMs mainly rely on deep structure, accuracy measures the correctness linking deep structure to output. In contrast, ADCE assesses the bidirectional relationship between deep structure to outputs, offering a more comprehensive evaluation. Specifically, we demonstrate that ADCE integrates the *probability of sufficiency* (PS) and *probability of necessity* (PN) (Pearl et al., 2000). For two boolean $X \in \{0, 1\}$ and $Y \in \{0, 1\}$, PS ($\delta_{\text{PS}}$) and PN ($\delta_{\text{PN}}$) measure how likely $X = 1$ causes $Y = 1$ given $X = 0, Y = 0$, and how likely $X = 0$ prevented $Y = 1$ given $X = 1, Y = 1$, respectively. In other words, PS assesses if $X = 1$ is sufficient to cause $Y = 1$, establishing a sufficient condition $X \Rightarrow Y$, while PN evaluates if $X = 1$ is necessary for $Y = 1$ to occur, determining a necessary condition $Y \Rightarrow X$. Theorem 1 demonstrates ADCE is a weighted combination of PS and PN, thereby capturing the bidirectional relationship between the sufficiency and necessity of deep structure changes on output variations.

**Theorem 1.** *(ADCE as a Combination of PN and PS) Let $T$ be the treatment variable in Equation 2 and $\hat{Y}$ the outcome of the indicator function in Equation 5. Assume $\hat{Y}$ is monotonic with respect to $T$, for ADCE, it holds that:*

$$\delta_{\text{ADCE}} = \frac{\alpha}{2} \cdot \delta_{\text{PS}} + \frac{\beta}{2} \cdot \delta_{\text{PN}} \tag{7}$$

*where $\alpha := \mathbb{P}(\hat{Y} = 1 | T = 1, s(T = 1))$, $\beta := \mathbb{P}(\hat{Y} = 0 | T = 0, s(T = 0))$.*

Theorem 1 demonstrates that ADCE quantifies the probability that modifications in deep structure are both necessary and sufficient for output variations. That is, ADCE measures the likelihood that deep structure alterations are the sole pathway leading observed changes in output. More introductions on PS and PN, along with detailed proof of Theorem 1 are in Appendix C.2.

**Output Depends on Surface Structure.** When models' outputs mainly depend on surface structure, e.g., spurious correlations, conventional accuracy metrics can be misleading (Ribeiro et al., 2016; Beery et al., 2018; Hashimoto et al., 2018; Duchi et al., 2019). For example, in sentiment classification tasks (Borkan et al., 2019; Koh et al., 2021), spurious correlations between identity and toxicity can lead models to misclassify texts containing identity information as toxic. While accuracy metrics based on these surface structure (e.g., identity information) might suggest high performance, they tend to overestimate the actual efficacy of the model. ADCE mitigates this by considering both sufficiency (identity information leading to toxicity) and necessity (toxicity not always implying identity information). This approach mitigates overreliance on spurious high-correlation paths from identity to toxicity, thus preventing performance overestimation. In Section 4.5, we empirically demonstrate that as the level of spurious correlation increases, accuracy remains misleadingly high, whereas ADCE declines. This demonstrates ADCE's superior ability to reflect a model's reliance on deep structure, particularly in scenarios dominated by spurious correlations.

## 4 EXPERIMENTS

In this section, we experimentally explore three critical questions: (1) **Deep structure comprehension in LLMs**: Do LLMs process questions through an understanding of the deep structure of problems? We analyze this using the proposed ADCE in Section 4.2. (2) **Prerequisite of deep structure comprehension**: What prerequisite enables LLMs to utilize deep structure in their responses? Insights into this question are discussed in Section 4.3? (3) **Comparative influence of deep and surface structures**: Which has a stronger causal effect on the outputs of LLMs – deep or surface structures? These investigations detailed in Section 4.4 collectively address the queries raised in Section 1, assessing whether LLMs are deep thinkers or merely surface structure learners. Additionally, to further support Section 3.4, we evaluate whether ADCE assesses core semantic understanding more reliably than accuracy under spurious correlations (in Section 4.5).

### 4.1 SETUP

**Dataset Evaluation and Intervention.** We employ five popular benchmarks across mathematics, logic, and commonsense knowledge. For mathematics, we consider 2-Digit Multiplication task

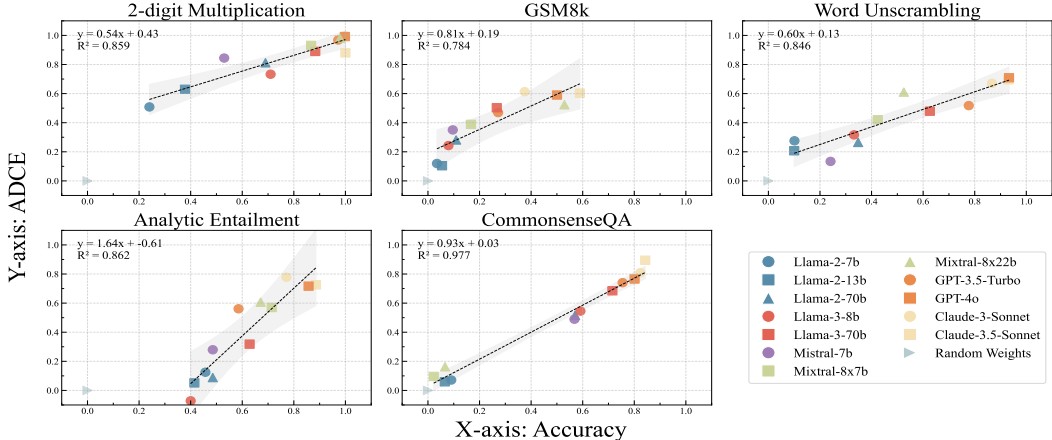

Figure 5: Deep structure understanding in LLMs via ADCE. Positive ADCE demonstrate the existence of direct causal effect of deep structure on outcomes, increasing with model scale and accuracy. Accuracy-DCE slopes vary across tasks, with steeper slopes indicating higher task complexity and greater reliance on various deep structure comprehension ability.

(bench authors, 2023) and GSM8k (Cobbe et al., 2021) for multi-step mathematical problems. Logical reasoning tasks include Word Unscrambling (bench authors, 2023), which requires unscrambling given letters to form an English word for implicit reasoning, and the binary Analytic Entailment task (bench authors, 2023) for linguistic entailment. Commonsense knowledge benchmarks include CommonsenseQA (Talmor et al., 2018), a multiple-choice task covering daily life knowledge.

Considering the diversity of experimental data, we explore various intervention strategies. Specifically, we use the *Mask* strategy for 2-Digit Multiplication, GSM8k and Word Unscrambling, which have key words representing core semantics. For Analytic Entailment and CommonsenseQA, with diverse presentation formats and less evident deep structure, we apply the *Rephrase* strategy. Appendix F.1 includes intervention examples and sample sizes of evaluated datasets after intervention.

**Models and Baselines.** We test 12 leading models from four LLM families: Llama (Llama-2-7b, Llama-2-13b, Llama-2-70b, Llama-3-8b, Llama-3-70b) (Touvron et al., 2023b; Dubey et al., 2024), Mistral (Mistral-7b, Mixtral-8x7b, Mixtral-8x22b) (Jiang et al., 2023; 2024a), GPT (GPT-3.5-Turbo, GPT-4o) (Achiam et al., 2023), and Claude (Claude-3-Sonnet, Claude-3.5-Sonnet) (Anthropic, 2024). Among them, Llama and Mistral families are open-source, while GPT and Claude are closed-source with inaccessible weights and architectures. A randomly weighted Llama-3-70b serves as a baseline denoting no direct causal effect between deep structure and outputs. Comparing its ADCE with other models evaluates our estimation method's effectiveness.

## 4.2 Deep Structure Comprehension Capability of LLMs

Figure 5 illustrates the relationship between accuracy and ADCE for 12 LLMs across five tasks. Notably, the ADCE for most models consistently remains positive, in stark contrast to the zero ADCE observed in the random weight baseline[1]. Positive ADCE values suggest that intervening deep structure causes LLMs to deviate from correct answers on previously solved problems, highlighting the models' reliance on deep structure for accurate problem-solving. This finding underscores that most LLMs possess deep structure understanding ability beyond surface structure.

Furthermore, comparing models within the same series (e.g., Llama-2, Llama-3, Mixtral), we observe that both accuracy and ADCE increase with model scale. A strong linear correlation emerges between accuracy and ADCE, with high $R^2 > 0.7$ indicating a good fit to the linear model. This suggests that models with higher accuracy exhibit greater dependence on deep structure for outputs.

Finally, slope $\beta$ of the accuracy-ADCE regression in Figure 5 quantifies the increase in deep structure understanding required per unit accuracy increase. Tasks like two-digit multiplication and word unscrambling show smaller $\beta$, indicating less deep structure comprehension needed for accuracy gains. GSM8k, Analytic Entailment and CommonsenseQA have higher $\beta$, emphasizing deep structure importance for accuracy. Variations in $\beta$ across tasks reflects underlying task complexity.

---

[1]Both Accuracy and ADCE of the random weight baseline are zero, indicating that this model neither comprehends problems nor makes random guesses. Outputs from the baseline are shown in Appendix F.2.

Low-$\beta$ tasks (e.g., 2-Digit Multiplication, Word Unscrambling) have fixed formats and single-skill requirements, needing small deep structure understanding for improvement. High-$\beta$ tasks (e.g., GSM8k, Analytic Entailment, CommonsenseQA) involve multi-step reasoning, diverse logical relationships and broad knowledge, demanding varied deep structure comprehension for accuracy gains.

### 4.3 THE PREREQUISITE OF DEEP STRUCTURE COMPREHENSION CAPABILITY

In Figure 5, certain LLMs, such as Llama-3-8b on Analytic Entailment, show minimal causal effects of deep structures on model output characterized by negative ADCE. This anomaly, where twisting deep structure improves accuracy, prompts an investigation into the specific conditions under which LLMs fail to comprehend deep structure across different tasks.

To investigate LLMs' failure, we explore the potential prerequisites for deep structure comprehension with positive ADCE. Inspired by previous work (Zečević et al., 2023; Jin et al., 2023), which proposes that the causality exhibited in LLMs often mirrors task-relevant knowledge embeded in their training data, we hypothesize that the absence of deep structure comprehension might indicate either unactivated or absent relevant knowledge in the training data. This theory proposes that missing replicable facts could hinder deep structure comprehension. To test this hypothesis, we employ supervised fine-tuning (SFT) to potentially activate task-specific knowledge (Gekhman et al., 2024; Allen-Zhu & Li, 2023; Zhou et al., 2024)[2]. Specifically, we fine-tune Llama-3-8b on Analytic Entailment and com-

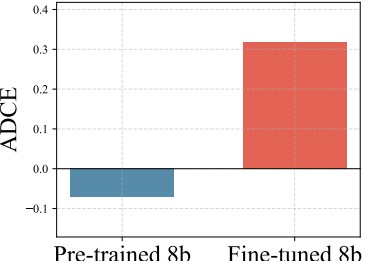

Figure 6: ADCE pre- and post- SFT. SFT activates entailment knowledge, enabling the model to exhibit deep structure causal effects on outcomes, as captured by proposed ADCE.

pare its ADCE before and after SFT. Figure 6 clearly illustrates an improvement in ADCE pre- and post-SFT, supporting that the ability to comprehend deep structures may rely on activating task relevant facts within the training data. Our findings also suggest that ADCE is effective for detecting such changes in comprehension pre- and post-activation. Further details on fine-tuning process are provided in Appendix G.

### 4.4 DEEP VS. SURFACE: A COMPARISON OF LLMS' COMPREHENSION ABILITY

After analyzing LLMs' deep structure comprehension and its potential sources, we extend our investigation to assess the reliance of LLMs on deep v.s. surface structures. This comparison aims to determine whether LLMs are deep thinkers or merely surface structure learners. We utilize ADCE in Equation 5 to measure the direct causal effect of deep structure, and an AICE, also specified in Equation 5, to quantify the indirect causal effect of surface structure while keeping deep structure constant. Figure 7 shows these comparisons, presenting ADCE as $\delta_{\text{ADCE}}$ and AICE as $\delta_{\text{AICE}}$. Our analysis reveals that closed-source models (e.g., GPT, Claude) primarily rely on deep structure, while open-source models (e.g., Llama) are more sensitive to surface structure. However, this sensitivity gradually decreases as model size increases, suggesting larger LLMs is more dependent on deep structure for answering. This analysis indicates that the tested closed-source models are not surface structure learners, as their responses rely more on deep structure. For the evaluated open-source LLMs, the dependency on surface structure tends to diminish as model scale increases.

### 4.5 ADCE VS. ACCURACY: CASE STUDY ON SPURIOUS CORRELATION

This section highlights the superiority of ADCE over traditional accuracy in measuring model reliance on deep structure, particularly in scenarios involving spurious correlations. Leveraging Civil-Comments (Borkan et al., 2019; Koh et al., 2021), a popular dataset for spurious correlation analysis, we manipulate the proportions of majority (spurious) and minority (non-spurious) group representations to construct training sets with differing degrees of spurious correlations. We then fine-tune

---

[2]Given the diversity of LLMs' training data (Dubey et al., 2024), we lean towards the view that relevant knowledge is not activated rather than absent from the training data.

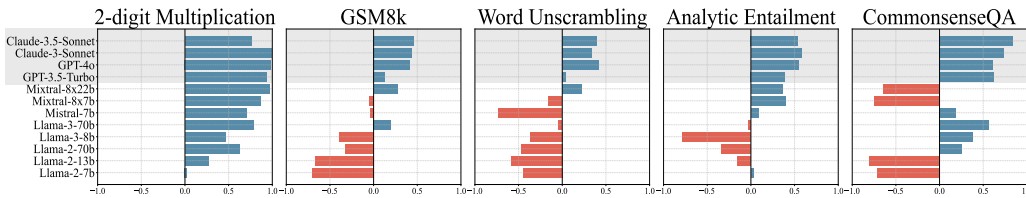

Figure 7: Comparing deep vs. surface structure. $\delta_{\text{ADCE}}$ represents ADCE of deep structure on output, while $\delta_{\text{AICE}}$ denotes AICE of surface structure on output. Closed-source models exhibit a greater reliance on deep structure for outputs. Open-source models (e.g. LLama-2) are more sensitive to surface structure; however, as model scale increases, this sensitivity is mitigated.

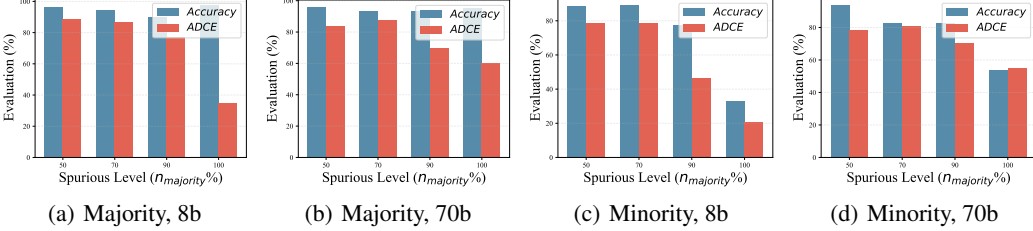

| (a) Majority, 8b | (b) Majority, 70b | (c) Minority, 8b | (d) Minority, 70b |

Figure 8: Spurious correlation results in LLama-3. In majority groups with spurious correlations, increasing correlation levels lead to high accuracy but declining ADCE. In minority groups without spurious correlations, accuracy and ADCE trends align. ADCE better reflects the model's reliance on spurious attributes over core semantics in spurious conditions, compared to accuracy.

Llama-3 using these specially prepared datasets. The subsequent evaluation involves comparing the model's accuracy and ADCE on the majority and minority group test sets, as depicted in Figure 8.

As the level of spurious correlations increases in the majority group, LLMs maintain high accuracy in the majority group, misleadingly predicting based on spurious attributes (i.e., identity information). Conversely, ADCE decreases, revealing the model's shift towards surface (spurious) structures over deep structure (i.e., core semantics). In contrast, in the minority group without spurious correlations, both accuracy and ADCE show consistent trends. This supports the argument in Section 3.4 that, in the presence of spurious correlations, ADCE provides a better measure of the model's reliance on deep structure compared to accuracy, without being artificially inflated by spurious attributes. More details on dataset construction and fine-tuning are presented in Appendix H.

## 5 RELATED WORK

Our related work primarily addresses the ongoing debate regarding LLMs' ability to comprehend deep and surface structure. Existing research has predominantly focused on LLMs' sensitivity to surface structure by modifying superficial patterns, such as substituting celebrity names, introducing misleading contexts (Jiang et al., 2024b; González & Nori, 2024), or altering the order of independent statements and options (Jiang et al., 2024b; Hooda et al., 2024; Turpin et al., 2024). These studies observe LLMs' lack of robustness through token-level and sentence-level interventions without altering core semantics, suggesting that LLMs' success relies heavily on recognizing surface structure. More aligned with our work, bench authors (2023) attempted a systematic analysis of the differences between in-context learning (ICL) and instruction-tuning (IT) in LLMs' understanding of domain knowledge in mathematical problems. They found that ICL better helps LLMs distinguish between deep and surface structure. These works inspire our research, which is more comprehensive and widely applicable to analyze LLMs' capacity for understanding deep and surface structure.

## 6 CONCLUSION

This paper investigate LLMs' comprehension abilities of deep and surface structures, proposing ADCE and AICE for quantification based on causal mediation analysis. ADCE analyses reveal LLMs' deep structure understanding across multiple tasks, potentially from activated task-specific knowledge in the training data. The comparison between ADCE and AICE reveals that closed-source LLMs comprehend deep structure better, while open-source LLMs exhibit higher surface sensitivity, which decreases as model scale increases. We demonstrate ADCE's superiority over accuracy in reflecting bidirectional deep structure-output relationships. This work hopes to provide new insights into LLMs' comprehension ability and offer novel methods for LLMs evaluation.

ACKNOWLEDGEMENTS

The authors would like to thank the anonymous reviewers and area chair for their helpful comments. This work is supported by the Shanghai Artificial Intelligence Laboratory. Difan Zou acknowledges the support from NSFC 62306252, Hong Kong ECS award 27309624, Guangdong NSF 2024A1515012444, and the central fund from HKU IDS.

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

## A   MORE EXAMPLES OF SURFACE AND DEEP STRUCTURE

In this section, we will provide more examples to illustrate the deep structure (core semantics) and surface structure (surface forms) of different inputs. Table 1 lists examples of 2-digit multiplication (bench authors, 2023). We then present the deep and surface semantics for the remaining four tasks described in Section 4.1.

- Word Unscrambling (bench authors, 2023): both Word Unscrambling task and 2-Digit Multiplication task have unified question templates and key tokens that reflect the core semantics. In Word Unscrambling, the question template is typically *The word X is a scrambled version of the English word*, where *X* is the scrambled word, such as *ofr* (a scrambled version of *for*). The key token reflecting the core semantics is *X*. Changes in surface structure, such as rephrasing the question to *How can the scrambled letters ofr be rearranged to form a valid English word?*, do not alter the answer to the problem.
- GSM8k (Cobbe et al., 2021): GSM8k is a dataset of multi-step reasoning elementary math problems with diverse question formats. For example: *A robe takes 2 bolts of blue fiber and half that much white fiber. How many bolts in total does it take?* The key tokens representing core semantics are numbers, quantifiers, etc. (e.g., *2*, *half*). Changing the surface structure, such as using symbolic notation, does not alter the problem's essence:

$$X = 2, \quad Y = X/2, \quad X + Y =?$$

  Where $X$ is blue fiber amount, $Y$ is white fiber amount, and ? is the total.
- Analytic Entailment (bench authors, 2023): Analytic Entailment is a task of determining logical relationships between sentences. The question format varies, for example: *Lina met two nurses.Lina met at least one woman.* The deep structure in Analytic Entailment is manifested in logical relationships and semantic inference, lacking uniform key tokens for core semantics. Altering the surface structure, such as: *Lina met two female nurses. Lina did not meet at least one woman.* does not change the nature of the task.
- CommonsenseQA (Talmor et al., 2018): CommonsenseQA, like Analytic Entailment, lacks a uniform question template. For example: *A revolving door is convenient for two direction travel, but it also serves as a security measure at a what?*. Its deep structure stems from understanding the question and context, without specific key tokens representing core semantics. Altering the surface structure, such as:*A revolving door is commonly used for easy entry and exit, but it also serves as a secure barrier between the outside and inside at a what?* does not change the answer, as the core concept remains intact.

## B   THE CAUSAL MEDIATION ANALYSIS

Causal Mediation Analysis (CMA) is a statistical method used to explain how an independent variable affects a dependent variable through one or more mediating variables (Baron & Kenny, 1986; Imai et al., 2010a; Coffman et al., 2021). This analytical approach is widely applied in many fields, such as psychology, sociology, and epidemiology (MacKinnon, 2012; Richiardi et al., 2013; Walters, 2018). Traditional mediation analysis is primarily quantifying mediation effects by comparing total (TE), direct (DCE), and indirect (ICE) causal effects (Rubin, 1974; Bollen & Davis, 2009; VanderWeele, 2009).

CMA places traditional mediation analysis within the potential outcomes framework (Rubin, 2005), using counterfactual reasoning to define and estimate causal effects (Pearl, 2001). This approach not only handles more complex mediation models but also better addresses confounding factors and sensitivity analyses (Imai et al., 2010a). A typical CMA framework comprises a treatment ($A$), a mediator ($M$), and an outcome ($Y$). Both $A$ and $M$ are observable variables that simultaneously influence $Y$. The primary objective of causal mediation analysis is to assess the causal effect of $A$ on $Y$ while isolating the influence of $M$ as illustrated in Figure 9.

In recent years, causal mediation analysis has also been widely applied in machine learning and artificial intelligence, providing new perspectives for explaining model decision processes and fairness assessments (Zhang & Bareinboim, 2018; Nabi & Shpitser, 2018).

It is important to emphasize that CMA is frequently applied to the traditional mediation model ($x \rightarrow z \rightarrow y$ and $x \rightarrow y$). Instead, we employ a variant of the classic causal mediation model

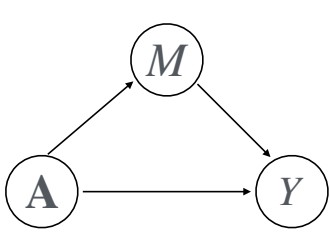
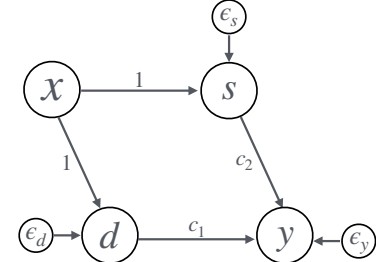

Figure 9: Typical mediation analysis graph with treatment $(A)$, mediator $(M)$ and outcome $(Y)$.

Figure 10: The Causal Graph of Synthetic Data which shares an identical causal graph as the interested intrested causal graph in Figure 3.

known as the Parallel Multiple Mediator Model (Preacher & Hayes, 2008; Bolin, 2014; VanderWeele & Vansteelandt, 2014). In our model, the deep structure $(d)$ and surface structure $(s)$ serve as two parallel mediators for the input x. The specific causal paths can be represented as $\boldsymbol{x} \to d \to Y$ and $\boldsymbol{x} \to s \to Y$.

Despite structural differences, our parallel multiple mediator model aligns with traditional mediation models in key aspects. Like classic mediation models, we also can decompose the total causal effect (TE: $\boldsymbol{x} \to Y$) into two parallel pathways: a direct causal effect (DCE: $\boldsymbol{x} \to d \to Y$) through our variable of interest (deep structure $d$), and an indirect causal effect (ICE: $\boldsymbol{x} \to s \to Y$) through the mediator (surface structure $s$). This decomposition mirrors the $x \to y$ and $x \to z \to y$ paths in traditional models and ensures that the relationship between TE, ICE, and DCE in Equation 3 holds. Additionally, our model satisfies key assumptions of causal mediation analysis which will be discussed in Appendix Appendix B.1. This fundamental consistency enables the application of established causal mediation methods to our model.

## B.1 ASSUMPTIONS IN CAUSAL MEDIATION ANALYSIS

To empoly thecausal mediation analysis, there are three positivity, consistency, and sequential ignorability need to be satisfied (Rubin, 1974; VanderWeele & Vansteelandt, 2009; Cole & Frangakis, 2009; Coffman et al., 2021; Nguyen et al., 2022; Qin, 2024).

**Positivity Assumption.** This assumption ensures that for all possible combinations of conditions, we can observe samples with non-zero probability, thereby allowing reliable estimation of causal effects. That is

**Assumption 1.** *(Positivity Assumption) For treatment $(A)$, mediator $(M)$, and an outcome $(Y)$ in Figure 9, it holds that:*

- *For the treatment variable $A$:*
$$\mathbb{P}(A = a) > 0, \quad \forall a \in \mathcal{A},$$
  *where $\mathcal{A}$ is the set of all possible values of $A$.*
- *For the mediator variable $M$:*
$$\mathbb{P}(M = m | A = a) > 0, \quad \forall m \in \mathcal{M}, a \in \mathcal{A}$$
  *where $\mathcal{M}$ is the set of all possible values of $M$.*
- *For the outcome variable $Y$:*
$$\mathbb{P}(Y = y | A = a, M = m) > 0, \quad \forall y \in \mathcal{Y}, a \in \mathcal{A}, m \in \mathcal{M}$$
  *where $\mathcal{Y}$ is the set of all possible values of $Y$.*

The positivity assumption is satisfied in our causal model. While as depicted in Figure 3, the intervention on the deep structure $d$ invariably induces a change in the surface structure $s$, for any given $d$, there exists a non-zero probability of observing each possible value of $s$ within the set $\mathcal{S}(d)$, where $\mathcal{S}(d)$ represents the range of $s$ values consistent with $d$. Thus, the essence of the positivity assumption—enabling causal inference for all structurally possible scenarios—is maintained, allowing for valid causal analysis within the model's defined constraints.

**Consistency Assumption.** The consistency assumption states that:When the treatment variable matches the theory potential treatment, the observed outcome in experiments should equal the potential outcome theoretically. Similarly, when the treatment variable matches, the observed mediator value in experiments should equal the potential mediator value theoretically. That is

**Assumption 2.** *(Consistency Assumption) For treatment ($A$), mediator ($M$), and an outcome ($Y$) in Figure 9, for individual $i$, it holds that:*

$$Y_i(a, M_i(a)) = Y_i \quad when \quad A_i = a,$$

*where $Y_i(a, M_i(a))$ is the potential outcome for individual $i$ under treatment $a$ and the corresponding potential mediator value $M_i(a)$, $Y_i$ is the observed outcome for individual $i$.*

$$M_i(a) = M_i \quad when \quad A_i = a$$

*where $M_i(a)$ is the potential mediator value for individual $i$ under treatment $a$, $M_i$ is the observed mediator value for individual $i$, $A_i$ is the observed treatment for individual $i$.*

In our study, all relevant variables are encompassed in Figure 3, thus precluding the existence of unobserved factors that could influence the mediator or outcome variables. Consequently, the consistency assumption is satisfied.

**Sequential Ignorability Assumption** Sequential ignorability involves two assumptions: (a) Conditional on the observed pre-treatment covariates, the treatment is independent of all potential outcomes and mediator values; (b) Conditional on the observed treatment and pre-treatment covariates, the observed mediator is independent of all potential outcomes. That is

**Assumption 3.** *For treatment ($A$), mediator ($M$), and an outcome ($Y$) in Figure 9, for individual $i$, it holds that:*

$$(a) \quad \{Y_i(a', m), M_i(a)\} \perp\!\!\!\perp A_i, \quad \forall a, a', m$$

$$(b) \quad Y_i(a', m) \perp\!\!\!\perp M_i(a)|A_i = a, \quad \forall a, a', m$$

*where $\perp\!\!\!\perp$ denotes statistical independence. $Y_i(a', m)$ is the potential outcome for under treatment $a'$ and mediator value $m$, $M_i(a)$ is the potential mediator value for unit $i$ under treatment $a$ and $A_i$ is the treatment assignment for $i$.*

Figure 3 presents a comprehensive causal graph encompassing all relevant variables and their causal relationships in this study. This completeness ensures the absence of unmeasured confounders. Furthermore, the independence between deep structure and surface variables structure is explicitly established. The completeness and independence jointly facilitate the satisfaction of the Sequential Ignorability Assumption (Imai et al., 2010a).

## B.2 CAUSAL EFFECTS IN CAUSAL MEDIATION ANALYSIS

Then, we introduce important causal estimands in the CMA framework, which characterize the causal effects between different variables. Consider the relationships between treatment ($A$), mediator ($M$), and an outcome ($Y$), all of them binary variables with values 0 or 1. Depending on the different values of the treatment and mediator variables, the causal effects between them primarily include the following types (Robins & Greenland, 1992; Pearl, 2001; VanderWeele, 2013):

- **Total Effect (TE):**
$$TE = E[Y(A = 1, M(1)) - Y(A = 0, M(0))] \tag{8}$$
- **Total Direct Effect (TDE):**
$$TDE = E[Y(A = 1, M(1)) - Y(A = 0, M(1))] \tag{9}$$
- **Pure Indirect Effect (PIE):**
$$PIE = E[Y(A = 0, M(1)) - Y(A = 0, M(0))] \tag{10}$$

Here, $Y(A = a, M(a))$ represents the value of $Y$ when $A = a$ and $M$ takes the value it would have when $A = a$. The total effect (TE) can be decomposed into direct effect and indirect effect (Robins & Greenland, 1992; Pearl, 2001; VanderWeele, 2013), i.e.,

$$TE = TDE + PIE \tag{11}$$

ADCE in Eq. (5) emphasizes deep structure' direct effect on the outcome, controlling mediator $s$ at post-intervention state (i,e., $s(T = 1)$). This control is necessary as changes in $d$ inevitably affect $s$. Thus, with intervention $T = 1$, we can only fix $s$ at $s(T = 1)$ instead of $s(T = 0)$. ADCE characterized in Equation 5 is actually the Total Direct Effect (TDE), while ICE is in fact the Pure Indirect Effect (PIE). Their relationship satisfy Equation 11. For a more understandable notation, we use the simpler concepts of ADCE and ICE in the main text to replace TDE and PIE.

## C  PROBABILITY OF SUFFICIENCY, NECESSITY AND PROOF

### C.1  PROBABILITY OF SUFFICIENCY AND NECESSITY

For two variables $X$ and $Y$, a sufficient condition is expressed as if $X$, then $Y$ ($X \to Y$), implying that the occurrence of $X$ inevitably leads to $Y$. Conversely, a necessary condition is expressed as $Y$ only if $X$ ($Y \to X$), indicating that the occurrence of $Y$ presupposes the prior existence of $X$.

We interpret above concepts from the probabilistic perspective, the Probability of Necessity (PN) and the Probability of Sufficiency (PS) (Pearl et al., 2000). PN measures that quantifies the relationship between two boolean variables $X$ and $Y$, defined as $PN(x, y) := P(y'_{x'}|x, y)$. Here, $y'_{x'}$ represents the counterfactual value of $Y = y'$ had X been set to a different value $x'$. By conditioning on both $X = x$ and $Y = y$, this measure reflects the likelihood of observing a different outcome in the absence of the event $X = x$. On the other hand, PS is defined as $PS(x, y) := P(y_x|x', y')$, which measures the probability that $X = x$ results in $Y = y$.

Since PN and PS cannot be estimated through observational data unless $Y$ is monotonic with respect to $X$ (Tian & Pearl, 2000). Therefore, we assume monotonicity of $Y$ with respect to $X$ and express PN and PS in computable forms as follows (Tian & Pearl, 2000; González & Nori, 2024):

$$\delta_{\text{PN}} = \frac{\mathbb{P}(Y = y) - \mathbb{P}(Y = y|\text{do}(X = x'))}{\mathbb{P}(X = x, Y = y)}, \tag{12}$$

$$\delta_{\text{PS}} = \frac{\mathbb{P}(Y = y|\text{do}(X = x)) - \mathbb{P}(Y = y)}{\mathbb{P}(X = x', Y = y')}. \tag{13}$$

The monotonicity assumptions and equations provide the foundation for the proof of Theorem 1.

### C.2  THE PROOF DETAILS

In this section, we provide the proof details of Theorem 1.

**Theorem 2.** *(Restatement of Theorem 1) Let $T$ be the treatment variable in Equation 2 and $\hat{Y}$ the outcome of the indicator function in Equation 5. Assume $\hat{Y}$ is monotonic with respect to $T$, for DCE, it holds that:*

$$\delta_{\text{DCE}} = \frac{\alpha}{2} \cdot \delta_{\text{PS}} + \frac{\beta}{2} \cdot \delta_{\text{PN}} \tag{14}$$

*where $\alpha := \mathbb{P}(\hat{Y} = 1|T = 1, s(T = 1))$, $\beta := \mathbb{P}(\hat{Y} = 0|T = 0, s(T = 0))$.*

*Proof.* We first define two binary variables as: Let $T$ be the treatment variable in Equation 2

$$T = \begin{cases} 0 & \text{intervention alters } s_i, \text{ preserves } d_i \\ 1 & \text{intervention alters both } s_i \text{ and } d_i \end{cases}$$

and $\hat{Y}$ the outcome of the indicator function in Equation 5.

$$\hat{Y} = \begin{cases} 0 & \text{if } Y^{\text{post}} = Y^{\text{pre}} \\ 1 & \text{if } Y^{\text{post}} \neq Y^{\text{pre}} \end{cases}$$

where $Y^{\text{post}}$ is the potential outcome after intervention.

Following assumptions in (Tian & Pearl, 2000; González & Nori, 2024), if $\hat{Y}$ is monotonic with respect to $T$, then PN and PS can be computed and represented as follows:

$$\delta_{\text{PN}}(T = 0, \hat{Y} = 0) = \frac{\mathbb{P}(\hat{Y} = 0) - \mathbb{P}(\hat{Y} = 0|\text{do}(T = 1))}{\mathbb{P}(T = 0, \hat{Y} = 0)} = \frac{\mathbb{P}(\hat{Y} = 0) - \mathbb{P}(\hat{Y} = 0|T = 1)}{\mathbb{P}(T = 0, \hat{Y} = 0)},$$

$$\delta_{\text{PS}}(T = 0, \hat{Y} = 0) = \frac{\mathbb{P}(\hat{Y} = 0|\text{do}(T = 0)) - \mathbb{P}(\hat{Y} = 0)}{\mathbb{P}(T = 1, \hat{Y} = 1)} = \frac{\mathbb{P}(\hat{Y} = 0|T = 0) - \mathbb{P}(\hat{Y} = 0)}{\mathbb{P}(T = 1, \hat{Y} = 1)}.$$

Notably, since there is no confounders between $T$ and $\hat{Y}$, $\mathbb{P}(\hat{Y}|\text{do}(T = t)) = \mathbb{P}(\hat{Y} = 0|T = t)$ (Pearl et al., 2000; Srihari, 2021).

According to the causal graph with mediation in Figure 3, the intervention $T$ on inputs $\boldsymbol{x}$ directly determines the state of the surface structure $s$, i.e.,

- When $T = 1$, it necessarily leads to $s(T = 1)$;
- When $T = 0$, it necessarily leads to $s(T = 0)$.

Therefore, we have

$$
\begin{aligned}
\mathbb{P}(\hat{Y}|T = t, s(T = t)) &= \frac{\mathbb{P}(\hat{Y}, T = t, s(T = t))}{\mathbb{P}(T = t, s(T = t))} \\
&= \frac{\mathbb{P}(s(T = t)|\hat{Y}, T = t)}{\mathbb{P}(s(T = t)|T = t)} \frac{\mathbb{P}(\hat{Y}, T = t)}{\mathbb{P}(T = t)} \\
&= \mathbb{P}(\hat{Y}|T = t)
\end{aligned}
$$

Therefore, we can simplify the ADCE expression without explicitly including $s$, e.g., simplify $\mathbb{P}(\hat{Y} = 1|T = 1, s(T = 1))$ as $\mathbb{P}(\hat{Y} = 1|T = 1)$

Then, the ADCE in Equation 5 can be redefined as

$$
\begin{aligned}
\hat{\delta}_{\text{ADCE}} &= \mathbb{P}(\hat{Y} = 1|T = 1, s(T = 1)) - \mathbb{P}(\hat{Y} = 1|T = 0, s(T = 0)) \\
&= \mathbb{P}(\hat{Y} = 1|T = 1) - \mathbb{P}(\hat{Y} = 1|T = 0) \\
&= \mathbb{P}(\hat{Y} = 0|T = 0) - \mathbb{P}(\hat{Y} = 0|T = 1) \\
&= \delta_{\text{PS}}(T = 0, \hat{Y} = 0) \cdot \mathbb{P}(T = 1, \hat{Y} = 1) + \delta_{\text{PN}}(T = 0, \hat{Y} = 0) \cdot \mathbb{P}(T = 0, \hat{Y} = 0).
\end{aligned}
$$

With the experiment setup that $\mathbb{P}(T = 1) = \mathbb{P}(T = 0) = \frac{1}{2}$, we obtain

$$
\hat{\delta}_{\text{ADCE}} = \frac{\mathbb{P}(\hat{Y} = 1|T = 1)}{2} \cdot \delta_{\text{PS}} + \frac{\mathbb{P}(\hat{Y} = 0|T = 0)}{2} \cdot \delta_{\text{PN}}.
$$

Here, we omit $(T = 0, \hat{Y} = 0)$ in PS and PN terms for simplicity. $\qquad \square$

## D  THE ALGORITHM OF ADCE

Algorithm 1 provides the detailed algorithmic steps required to estimate ADCE, which includes the following: First, we perform initial inference on the full dataset to select samples with correct answers. Then, for these correctly answered samples, we apply interventions using two strategies: Masking and Rephrasing. Finally, we conduct a second round of inference on the intervened samples and calculate ADCE based on the inference results.

---

**Algorithm 1:** Approximated Direct Causal Effect (ADCE) Estimation in LLMs

---

**Input:** Dataset $\mathcal{D} = \{\boldsymbol{x}_i, y_i\}_{i=1}^n$, LLM $f_{\boldsymbol{\theta}}$, intervention strategy $\mathcal{I}$
**Output:** Estimated ADCE

1 **Stage 1:** Initial Inference on Full Data
2 $\mathcal{D}_c \leftarrow \{\boldsymbol{x}_i \in \mathcal{D} : f_{\boldsymbol{\theta}}(\boldsymbol{x}_i) = y_i\}$      // Collect correctly answered samples
3 $Y_{pre} \leftarrow f_{\boldsymbol{\theta}}(\mathcal{D}_c)$                           // Original Outcome
4 **Stage 2:** Generate Intervention Data (Alg. 2)
5 $\mathcal{D}_{T=1}, \mathcal{D}_{T=0} \leftarrow \mathcal{M}_{\mathcal{I}}(\mathcal{D}_c)$
6 **Stage 3:** Re-Inference on Intervention Data
7 **for** $i \in \{0, 1\}$ **do**
8     $Y(T = i, s(T = i)) \leftarrow f_{\boldsymbol{\theta}}(\mathcal{D}_{T=i})$ // Potential Outcomes for TE and AICE
9 **end**
10 **Stage 4:** Estimate ADCE via Equation 5
11 **return** Estimated ADCE

---

## E  EXPERIMENTS ON SYNTHETIC DATA

In this section, we validate our proposed framework using synthetic data where true causal effects can be calculated to evaluate the effectiveness of ADCE and AICE. We base our synthetic data on the simplified causal graph shown in Figure 3, which represents real scenarios. Our model considers

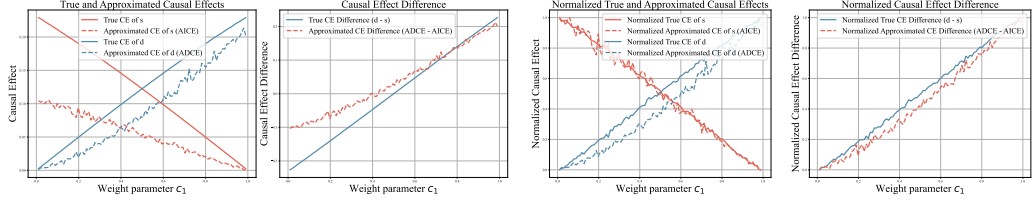

(a) Unnormalized Causal Effects         (b) Normalized Causal Effects

Figure 11: Comparison of True Causal Effects (True CE of $d$ and $s$) and Approximated Causal Effects (Approximated CE of $d$ and $s$ i.e., ADCE and AICE) on synthetic data. With known true causal effects, both the true and approximated causal effects of $d$ and $s$ on the model's output demonstrate consistent trends. The differences in causal effects between $d$ and $s$ also show similar patterns. After normalization, the true causal effects and approximated causal effects align more closely.

four key variables: input $\boldsymbol{x}$, deep structure $d$, surface structure $s$ and outputs $y$. The synthetic data we generate adheres to the causal graph presented in Figure 10 and follows the Structural Causal Models (SCM) (Pearl, 2009) described as follow.

$$x \sim \mathcal{N}(0,1), \quad d = x + \epsilon_d, \quad s = x + \epsilon_s. \tag{15}$$

$$y = \begin{cases} 1, & \text{if } \sigma(c_1 \cdot d + c_2 \cdot s + \epsilon_y) > 0.5 \\ 0, & \text{otherwise} \end{cases} \tag{16}$$

where we consider an independent small noise $\epsilon_d \sim \mathcal{N}(0, 0.25)$ and $\epsilon_s \sim \mathcal{N}(0, 0.25)$. And the independent noise $\epsilon_y \sim \mathcal{N}(0,1)$ and $\sigma(\cdot)$ is Sigmoid function. $c_1$ and $c_2$ are weight parameters for $d$ and $s$, respectively. Analogously, larger $c_1$ (or $c_2$) indicate more prominent deep (or surface) structure signals in inputs. Equations 15 and 16 are simplification of the true causal graph shown in Figure 3 which reduces $d$, $s$, and $\boldsymbol{x}$ to scalars and assumes they exhibit simple linear relationships. Despite simplification, this SCM retains the key causal relationships in Figure 3, where $x$'s effect on is mediated through two paths: $x \to d \to y$ and $x \to s \to y$. Then, we generate the training data and train a logistic regression $f$ with explicit functions and parameters, ensuring clear model's dependencies on $d$ and $s$ for outputs. Explicit functions and parameters enable direct computation of true causal effects for ADCE and AICE validation. Specially, we generate 100000 training samples for model $f$, defining true causal effects of $f$'s dependence on $d$ and $s$ as their respective average marginal effects (AMEs) (Schennach et al., 2007; Breen et al., 2018; Aguirregabiria & Carro, 2024). AMEs represent average output changes when only $d$ or $s$ increases by one unit. Via predicting on 10000 test samples, we compute (1) TE in Equation 5 by setting $d = 0$ and $s' = s + \epsilon_{s'}$ where $\epsilon_{s'} \sim \mathcal{N}(0, 0.25)$, (2) AICE in Equation 5 by setting $s = s'$ where we use the same $s'$ in TE and (3) ADCE in Equation 5 by calculating ADCE = TE − AICE.

Figure 11(a) shows how true causal effects of $s$ and $d$ on model output change as $d$'s weight $c_1$ increases. As $c_1$ rises, the logistic model's more dependent on deep structure for outputs with increased $d$'s true causal effect and decreased $s$'s true causal effect. The estimated versions, ADCE and AICE, follow similar trends, validating their effectiveness. Figure 11(a) also displays the difference between $d$ and $s$ causal effects. The estimated difference aligns with the true difference, supporting our comparative results in Section 4.4. Furthermore, true causal effects range from 0 to 0.25, while ADCE spans $[-1, 1]$, hindering direct comparisons. We normalize both causal effects to $[0, 1]$ for fair comparison in Figure 11(b). The normalized estimates align closely with true effects, with difference curves align more closely, further validating ADCE and AICE.

# F  DATASETS AND MODELS

## F.1  DETAILS OF GENERATING INTERVENTION DATASETS: METHOD AND DATA SIZE

### F.1.1  INTERVENTION METHOD

In this section, we first outline the detailed process for generating the intervention data required for computing TE and ICE in Algorithm 2.

---

**Algorithm 2:** Intervention Data Generation Method $\mathcal{M}$

---

**Input:** Correctly answered samples $\mathcal{D}_c = \{(\boldsymbol{x}_i, y_i)\}$, LLM $f_{\boldsymbol{\theta}}$, intervention strategy $\mathcal{I}$, and
  LLM agent $\mathcal{C}$

**Output:** Intervention datasets $\mathcal{D}_{T=1}, \mathcal{D}_{T=0}$

1 **for** $(\boldsymbol{x}, y) \in \mathcal{D}_c$ **do**
2    **if** $\mathcal{I} = Mask$ **then**                     // Generate $(T = 1, s(T = 1))$ data
3       $\boldsymbol{x}_{T=1} \leftarrow \text{MaskCoreSemantics}(\boldsymbol{x})$
4    **else**
5       $\boldsymbol{x}_{T=1} \leftarrow \text{RephraseByAgent}(\boldsymbol{x}, y, \mathcal{C}, \text{``Alter''})$
6    $\mathcal{D}_{T=1} \leftarrow \mathcal{D}_{T=1} \cup \{(\boldsymbol{x}_{T=1}, y)\}$
7    **if** $\mathcal{I} = Mask$ **then**                     // Generate $(T = 0, s(T = 0))$ data
8       tokens $\leftarrow \text{GetNonCoreSemanticTokens}(\boldsymbol{x})$
9       nearestTokens $\leftarrow \text{GetKNearestTokens}(\text{tokens}, \boldsymbol{x}_{T=1}, k)$
10      $\boldsymbol{x}_{T=0} \leftarrow \text{MaskTokens}(\boldsymbol{x}, \text{nearestTokens})$
11    **else**
12      $\boldsymbol{x}_{T=0} \leftarrow \text{RephraseByAgent}(\boldsymbol{x}_{T=1}, y, \mathcal{C}, \text{``Preserve''})$
13    $\mathcal{D}_{T=0} \leftarrow \mathcal{D}_{T=0} \cup \{(\boldsymbol{x}_{T=0}, y)\}$
14 **return** $\mathcal{D}_{T=1}, \mathcal{D}_{T=0}$

---

We then provide more details on the intervention data generation according to different strategies.

**The *Mask* Strategy.** For 2-Digit Multiplication, GSM8k, and Word Unscrambling tasks, we employ the *Mask* strategy to construct the corresponding intervention data. We establish specific intervention word pool for each task, where intervening on words specified in these words results in disruption of the core semantics (i.e., deep structure). The post-intervention samples are used to calculate TE in Equation 5. Conversely, intervening on words outside these rules only causes surface structure changes, and the resulting samples are used to compute AICE in Equation 5. Intervening on words specified in the intervention word pool leads to changes in the deep structure of inputs. In our experiments, we select one word at a time from the pool of candidate words and replace it with <Mask>. For ICE, when masking words outside the intervention word pool, we consider the nearest non-semantic word for masking based on the word masked in TE, i.e., $k = 1$.

- 2-Digit Multiplication: We apply the *Mask* strategy to all *numerical digits* and the multiplication operator (*times*) to induce changes in the core semantic structure. Conversely, masking any tokens other than digits and the multiplication operator is regarded as altering only the surface structure.
- GSM8k: For the GSM8k task, we define an intervention word pool that, when masked, alters the core semantic structure. This pool encompasses all *numerical digits* and the following lexical items representing mathematical operations and other numerical representations: {*zero, one, two, three, four, five, six, seven, eight, nine, ten, eleven, twelve, thirteen, fourteen, fifteen, sixteen, seventeen, eighteen, nineteen, twenty, thirty, forty, fifty, sixty, seventy, eighty, ninety, hundred, thousand, million, billion, times, minus, plus, divided, multiplied, dozen, twice*}. The intervention strategy is designed to guarantee that every instance in the dataset undergoes a significant semantic transformation through the masking of one critical term from the given intervention word pool.
- Word Unscrambling: For the Word Unscrambling task, the question template is consistently structured as *The word X is a scrambled version of the English word*, where X represents the scrambled word (e.g., X=*hte* for *the*, X=*adn* for *and*). We determine that masking the third position word (i.e., X) alters the core semantic structure. Correspondingly, when $k = 1$, masking either *word* or *is* only modifies the surface structure.

**The *Rephrase* Strategy**. We select `claude-3-5-sonnet` model as the LLM agent for paraphrase generation and define a set of templates with different utilities. Note that these templates can be customized for different tasks, which contribute to the versatility of the proposed intervention framework in intervening natural language datasets. The detailed rephrasing framework is depicted in Algorithm 3, which generally includes three steps: paraphrase generation, generation check, and feedback saving. First, according to the rephrasing target $\mathcal{T}$, the framework constructs prompt based on the appropriate template from Table 4. The prompt will then be sent to the LLM agent for rephrasing, with paraphrase $\boldsymbol{x}'$ as the output. Next, we ask the agent to predict the label of $\boldsymbol{x}'$. If the

prediction matches the expectation, we break and return the generated text. Otherwise, we record the generated text and send feedback to LLM for the next generation. The whole process will be repeated until the agent generate the desired paraphrase.[3] The examples of generated paraphrases are listed in Table 3.

Table 3: Examples of generated paraphrases of CommonsenseQA and Analytic Entaiment datasets using Claude-3.5-Sonnet API. We carefully design our intervention strategy to ensure that $s(T = 1)$ and $s(T = 0)$ are as similar as possible, in order to satisfy the approximation.

| Dataset | State | Text |
|---|---|---|
| | Origin | What do people aim to do at work? A: complete job |
| | $T = 1, s(T = 1)$ | What do people primarily aim to do during work breaks? A: talk to each other |
| | $T = 0, s(T = 0)$ | What do people primarily aim to do during overtime hours? A: complete job |
| | Origin | What do people typically do while playing guitar? A: singing |
| | $T = 1, s(T = 1)$ | What do people typically avoid doing while playing guitar? A: cry |
| | $T = 0, s(T = 0)$ | What do people typically do simultaneously while playing guitar? A: singing |
| | Origin | After he got hired he hoped for success at his what? A: new job |
| CommonsenseQA | $T = 1, s(T = 1)$ | After he got hired as a volunteer, he hoped for success at his what? A: vocation |
| | $T = 0, s(T = 0)$ | After he got hired as an employee, he hoped for success at his what? A: new job |
| | Origin | Where would a person be doing when having to wait their turn? A: stand in line |
| | $T = 1, s(T = 1)$ | Where would a person likely be if they didn't have to wait their turn? A: sing |
| | $T = 0, s(T = 0)$ | Where would a person likely be if they had to wait their turn? A: stand in line |
| | Origin | Where is a doormat likely to be in front of? A: front door |
| | $T = 1, s(T = 1)$ | Where is a doormat least likely to be placed in front of? A: facade |
| | $T = 0, s(T = 0)$ | Where is a doormat most likely to be placed in front of? A: front door |
| | Origin | Sarah has a pet. So Sarah has a dog. A: no-entailment |
| | $T = 1, s(T = 1)$ | Sarah has a dog. So Sarah has a pet. A: entailment |
| | $T = 0, s(T = 0)$ | Sarah has a dog. Sarah has a car. A: no-entailment |
| | Origin | Wendy has zero kids. So Wendy has a number of kids. A: no-entailment |
| | $T = 1, s(T = 1)$ | Wendy has zero kids. So Wendy is childless. A: entailment |
| | $T = 0, s(T = 0)$ | Wendy has zero kids. So Wendy is not childless. A: no-entailment |
| | Origin | Richard yelled at Ethan. Therefore Richard yelled. A: entailment |
| Analytic Entailment | $T = 1, s(T = 1)$ | Richard yelled at Ethan. Therefore, Ethan yelled. A: no-entailment |
| | $T = 0, s(T = 0)$ | Richard yelled at Ethan. Therefore, Ethan was yelled at. A: entailment |
| | Origin | Tom is George's grandfather. So, George is a descendant of Tom's. A: entailment |
| | $T = 1, s(T = 1)$ | Tom is George's grandfather. So, George looks up to Tom. A: no-entailment |
| | $T = 0, s(T = 0)$ | Tom is George's grandfather. So, George is Tom's grandson. A: entailment |
| | Origin | The tabletop is square. So, the tabletop is rectangular. A: entailment |
| | $T = 1, s(T = 1)$ | The tabletop is square. So, the tabletop is large. A: no-entailment |
| | $T = 0, s(T = 0)$ | The tabletop is square and large. So, the tabletop is large. A: entailment |

### F.1.2 INTERVENTION DATA SIZE

In this section, we introduce the sample sizes before and after intervention.

- 2-Digit Multiplication: For the two-digit multiplication problem, the original dataset comprised 1000 samples. Following Algorithm 2, we perform interventions on correctly answered samples with accuracy $\alpha$ for each LLM $f_\theta$. For each sample, we generate two intervention groups with *Mask* strategy: first synthesizing one sample with altered core semantics (deep structure), then based on this, synthesizing another with only surface structure changes. This process is repeated twice, resulting in 4 intervention samples per original sample: 2 with deep structure changes and 2 corresponding samples with only surface structure changes. In total, for LLM $f_\theta$, $4000\alpha$ intervention samples are generated (4 per original sample).
- GSM8k: For GSM8k, the original dataset consisted of 1319 samples. Following Algorithm 2, we conduct interventions on correctly answered samples for each LLM $f_\theta$ with accuracy $\alpha$. For each sample, we also generate two intervention groups with *Mask* strategy: first synthesizing one sample with altered core semantics (deep structure), then generating another with only surface structure changes based on this. This process is repeated twice, yielding 4 intervention samples

---

[3]In practice, we set the maximal iteration number as 10 to avoid prohibitive long context.

---

**Algorithm 3:** RephraseByAgent

---

**Input:** Text $x$, label $y$, rephrasing target $\mathcal{T}$, and LLM agent $\mathcal{C}$
**Output:** $x'$

1 **if** $\mathcal{T}$ = "*Alter*" **then**                      // Generate prompt for paraphrase
2     prompt ← Table 4.**Template 1**
3 **else**
4     prompt ← Table 4.**Template 2**
5 chatHistory = prompt.format($x$)          // Insert questions, options and the answer inside the placeholders
6 selfCheckFlag = False
7 **repeat**
8     $x' \leftarrow \mathcal{C}$(chatHistory);                           // Step 1:  Generation
9     predictionPrompt ← Table 4.**Template 3**
10     $y' \leftarrow \mathcal{C}$(predictionPrompt.format($x'$));                 // Step 2:  Self-check
11     **if** *($\mathcal{T}$ = "Alter" **and** $y' \neq y$) **or** ($\mathcal{T}$ = "Preserve" **and** $y' = y$)* **then**
12       selfCheckFlag ← True
13     **else**
14       chatHistory ← chatHistory + $x'$
15       chatHistory ← chatHistory + Table 4.**Template 4** ;        // Step 3:  Feedback
16 **until** *selfCheckFlag = True*;
17 **return** $x'$

---

per original sample: 2 with deep structure changes and 2 corresponding samples with only surface structure modifications. In total, for LLM $f_\theta$, $5276\alpha$ intervention samples are generated (4 per original sample).

- Word Unscrambling: For Word Unscrambling, we sample 1000 instances from the original full dataset. Following Algorithm 2, we conduct interventions on correctly answered samples for each LLM $f_\theta$ with accuracy $\alpha$. For each sample, we generate two intervention groups using the *Mask* Strategy: first synthesizing one sample with altered core semantics (deep structure), then generating another with only surface structure changes based on this. This process is performed once, yielding 2 intervention samples per original sample: 1 with deep structure changes and 1 with corresponding surface structure modifications. In total, for LLM $f_\theta$, $2000\alpha$ intervention samples are generated (2 per original sample).

- Analytic Entailment: For Analytic Entailment, the original dataset comprise 70 samples. Following Algorithm 2 and Algorithm 3, we conduct interventions on correctly answered samples for each LLM with accuracy $\alpha$. For each sample, we apply two intervention groups using the *Rephrase* Strategy: first synthesizing one sample with altered core semantics (deep structure), then generating another with only surface structure changes based on this. This process is repeated twice, yielding 4 intervention samples per original sample: 2 with deep structure changes and 2 with corresponding surface structure modifications. In total, for LLM $f_\theta$, $280\alpha$ intervention samples are generated (4 per original sample).

- CommonsenseQA: For CommonsenseQA, the original dataset contain 1221 samples. Following Algorithm 2, we conduct interventions on correctly answered samples for each LLM with accuracy $\alpha$. For each sample, we apply two intervention groups using the *Rephrase* Strategy: first synthesizing one sample with altered core semantics (deep structure), then generating another with only surface structure changes based on this. This process is repeated twice, yielding 4 intervention samples per original sample: 2 with deep structure changes and 2 with corresponding surface structure modifications. In total, for LLM $f_\theta$, $4884\alpha$ intervention samples are generated (4 per original sample).

## F.2 RANDOM WEIGHTED BASELINE

We employ `AutoModelForCausalLM.from_config` to load a new model with an model architecture identical to LLama-3-70b but with randomly initialized weights as our baseline. This

Table 4: Prompts for automatic causal interventions, where the text in `monospaced` font can be tailored to different tasks.

---

**[Template 1] Rephrase & Alter**
You are an expert in `natural language processing and commonsense reasoning`. Your task is to rephrase the given `commonsense` question, and then modify the paraphrase so that the modified question results in a different answer based on the provided options. The input will be in the form of a dictionary: {'Question':'question', 'Options':['option1', 'option2',...], 'Answer':'ans'}, where 'Question' is the original `commonsense` question, 'Options' are the candidate answers, and 'Answer' is the original correct answer. Output only the modified Question without any introductory phrases.
Here is the input: {'Question': `[QUESTION]`,'Options':`[OPTIONS]`,'Answer':`[ANSWER]`}. The modified question is:

---

**[Template 2] Rephrase & Preserve**
You are an expert in `natural language processing and commonsense reasoning`. Modify the keywords with minimal word changes in the 'Question' to ensure the given 'Answer' is the most fitting answer to the modified result among the 'Options'. The input is in the form of a dictionary: {'Question':'question', 'Options':['option1', 'option2', ...], 'Answer':'ans'}. Output only the modified Question without any introductory phrases.
Here is the input: {'Question': `[QUESTION]`,'Options':`[OPTIONS]`,'Answer':`[ANSWER]`}. The modified question is:

---

**[Template 3] Prediction**
You are an expert in `natural language processing and commonsense reasoning`. Below is a `commonsense` question along with some answer options. Choose the correct answer from these options. Your output should only be the answer enclosed in parenthesis, without any introductory phrases.
Question: `[QUESTION][OPTIONS]`
Among `[INDEX_OF_FIRST_OPT]` through `[INDEX_OF_LAST_OPT]`, the answer is

---

**[Template 4] Feedback**
The answer to the modified question is `different from` the original question. Please modify the question again. Output only the modified Question.

---

random baseline model is incapable of comprehending the task, let alone making random guesses. We provide examples of its output as follows:

# G  FINE-TUNING ON ANALYTIC ENTAILMENT DATASET

## G.1  SUPERVISED FINE-TUNING ON ANALYTIC ENTAILMENT DATASET

To fine-tune the llama-based models, we utilize the `llama-recipes` library[4] and train the models on a cloud server with 2 NVIDIA Tesla A100 GPUs with 80G memory of each. We employ LoRA (Hu et al., 2022) technique from the `peft` library[5] for memory-efficient training.

For Analytic Entailment dataset, we include the generated paraphrases for training and evaluation. For each question, we generate two sets of paraphrases as depicted in Appendix F.1, with each set include one $(T = 1, s(T = 1))$ sample and $(T = 0, s(T = 0))$ sample. Based on this, we expanded our dataset from 70 original samples to a total of 350 samples, with each set comprising one original sample and four corresponding paraphrases. We then divided these 70 sets for training and testing with a ratio of $6 : 4$. Consequently, we obtained a training set consisting of 210 samples derived from 42 original samples and a test set comprising 140 samples, which were derived from the intervention on 28 original samples.

We set the batch size to be 20 and set the learning rate to be 0.0003 for both `llama-3-8b` and `llama-3-70b`. For other parameters, we use the default value as defined in the official code from

---

[4]`https://github.com/meta-llama/llama-recipes`
[5]`https://huggingface.co/docs/peft`

Table 5: The baseline with random weights maintains the same architecture as LLama-3-70b but is incapable of comprehending specific problems, let alone making random guesses. Consequently, both its accuracy and ADCE are zero.

---

**[Task 1] 2-Digit Multiplication:**
Input: You are a helpful assistant for arithmetic reasoning. You are required to answer the following question and answer in the format of 'The answer is'. What is 50 times 20?
Output: MTatual novemberdxanime alqun =filestitutions Riveraixe Nga juego.Note

---

**[Task 2] GSM8k**
Input: You are a helpful assistant for arithmetic reasoning. You are required to answer the following question and answer in the format 'The answer is'. Kyle bought last year's best-selling book for \$19.50. This is with a 25% discount from the original price. What was the original price of the book?
Output: ulseries/sys tab ArapDAMAGEestyIILISE Mayboruev ance Signatureappeno

---

**[Task 3] Word Unscrambling**
Input: As a linguistics expert, you will be provided with scrambled letters and identify the correct English word that they can form. Answer in the format 'The unscrambled word is'. The word hte is a scrambled version of the English word
Output: Sellertheseevilervadirs estruct mparator]]);=¿ maxugas

---

**[Task 4] Analytic Entailment**
Input: Determine whether the following pairs of sentences embody an entailment relation or not. Some people ordered beverages. So no one ordered orange juice.
Output: shortly airline (bodyParservenue Rapids (slugotrab coeffvelocity

---

**[Task 5] CommonsenseQA**
Input:You are an expert in natural language processing and commonsense reasoning. Below is a commonsense question along with some answer options. Choose the correct answer from these options. Kyle bought last year's best-selling book for \$19.50. This is with a 25% discount from the original price. What was the original price of the book?
Output: ROSS Residents.radfrom processesSi nouvel Full)[PIE()" DVD=centeryyy

---

`llama-recipes` library. We train the models until convergence, and both `llama-3-8b` and `llama-3-70b` converge within 200 steps.

### G.2 MORE POST TRAINING STRATEGIES

In this section, we have expanded our analysis to include two additional post-training approaches: Instruction Fine-Tuning (IFT) (Wei et al., 2021) and Fine-Tuning with In-Context Learning (FTICL) (Anil et al., 2022). We've also analyzed the In-Context Learning (ICL) (Brown, 2020) method, due to its effectiveness in harnessing the models' inherent abilities to comprehend and produce responses, as well as its popularity within the NLP community. Following the experimental setting in Section 4.3, we also consider Llama-3-8b on the Analytic Entailment task. Specifically, for IFT, we augment each input text with the following template:

Table 6: The prompt for IFT. We consider the performance of LLama-3-8b on the Analytic Entailment task.

---

**Template for IFT**

As an expert in linguistic entailment, you will be provided with two sentences and determine if there is an entailment relationship between sentence 1 and sentence 2. An entailment relationship exists when the truth of sentence 1 guarantees the truth of sentence 2.

**Sentences:** `[INPUT]`

**Relation:** (entailment or no-entailment):

---

Here, `[INPUT]` will be replaced by the input text. In addition to the instructions used in IFT, for FTICL, we incorporate two examples with corresponding ground truth into the template:

Table 7: The prompt for FTICL. We consider the performance of LLama-3-8b on the Analytic Entailment task.

---

**Template for FTICL.**

---

As an expert in linguistic entailment, you will be provided with two sentences and determine if there is an entailment relationship between sentence 1 and sentence 2. An entailment relationship exists when the truth of sentence 1 guarantees the truth of sentence 2.

**Sentences:** [INPUT]

**Relation:** (entailment or no-entailment):

---

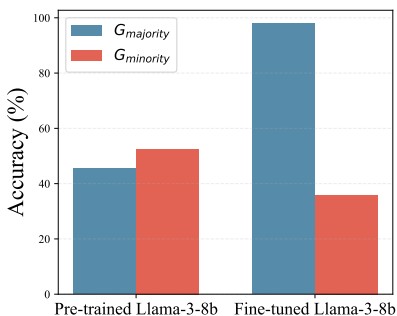

Figure 12: SFT on LLama-3-8b

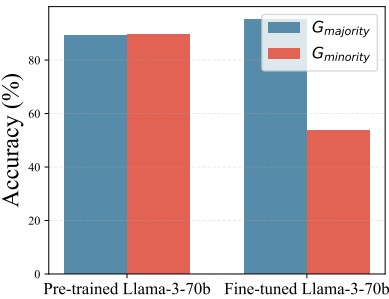

Figure 13: SFT on LLama-3-70b

Figure 14: Introducing spurious correlations into the initially unbiased LLama-3 series through fine-tuning, with spurious level $n_{\mathrm{majority}} = 100$

For ICL, we utilize the same sample template as in FTICL. The key difference is that ICL does not involve finetuning the models; instead, it employs this template solely for evaluation purposes. The results are provided below:

Table 8: Comparison of different metrics across various training stages. We consider the performance of LLama-3-8b on the Analytic Entailment task.

| Metric | Pre-training | SFT | IFT | FTICL | ICL |
|---|---|---|---|---|---|
| Accuracy | 0.457 | 0.743 | 0.800 | 0.786 | 0.771 |
| ADCE | -0.071 | 0.318 | 0.478 | 0.533 | 0.455 |

We find that various post-training strategies and ICL all lead to improvements in both model accuracy and deep structure understanding ability (ADCE). Moreover, FTICL and IFT, which consider both prompt engineering and parameter optimization, yield greater gains compared to SFT, which only focuses on parameter optimization, or ICL, which only utilizes prompts.

## H  EXPERIMENTAL DETAILS ON SPURIOUS CORRELATION

**Construction of Spurious Correlation Data.** We initially sample from Civilcomments to construct training datasets with varying degrees of spurious correlations. The sampling procedure selects 2500 extreme samples with toxicity probability $> 0.8$ and containing identity, assigning label 1 (toxic), and 2500 extreme samples with toxicity probability $< 0.2$, assigning label 0 (non-toxic) for the majority group with spurious correlations. For the minority group without spurious correlations, we select samples with toxicity probability $> 0.5$ and no identity, assigning label 1, and samples with toxicity probability $< 0.5$ and containing identity, assigning label 0. We adjust the proportion of the majority group while maintaining a total sample size of 4526. For instance, a 50% majority group implies 2263 samples each in the majority and minority groups. We consider four settings with increasingly spurious correlations level, where $n_{\mathrm{majority}}$ accounts for 50%, 70%, 90%, and 100% of the total samples. For the test data, after sampling the training set, we apply the same sampling rules to the remaining population. We select 200 samples each from the majority and minority groups

Table 9: Values of Accuracy, ADCE, and AICE for different noise levels $\eta$ on data with text noise.

| $\eta$ | Accuracy | ADCE | AICE |
|---|---|---|---|
| 0 | 0.710 | 0.733 | 0.264 |
| 0.2 | 0.497 | 0.681 | 0.319 |
| 0.5 | 0.201 | 0.550 | 0.448 |
| 0.7 | 0.093 | 0.438 | 0.556 |
| 0.9 | 0.031 | 0.444 | 0.556 |

Table 10: Values of Accuracy, ADCE, and AICE for different noise levels $\eta$ on data with label noise.

| $\eta$ | Accuracy | ADCE | AICE |
|---|---|---|---|
| 0 | 0.710 | 0.733 | 0.264 |
| 0.2 | 0.497 | 0.681 | 0.319 |
| 0.5 | 0.201 | 0.550 | 0.448 |
| 0.7 | 0.093 | 0.438 | 0.556 |
| 0.9 | 0.031 | 0.444 | 0.556 |

within this population. We then employ the rephrase method proposed in Algorithm 3 to construct intervention data for accuracy and DCE.

**Fine-tuning on Spurious Correlation Data.** We set the batch size to be 50, and set the learning rate to be 0.001 and 0.0003 for `llama-3-8b` and `llama-3-70b`, respectively. For other parameters, we use the default value as defined in the official code from `llama-recipes` library. We train the models until convergence. In all training cases, the models converge within 250 steps.

# I  EXPERIMENTS ON NOISY DATA

In this section, we extend our experiments to NLP tasks with noisy data. We consider two scenarios: text noise (Belinkov & Bisk, 2017; Karpukhin et al., 2019; Wei & Zou, 2019) and label noise (Garg et al., 2021; Wu et al., 2023). For demonstration, we use the 2-digit Multiplication dataset and LLama-3-8b model as an example.

**Text Noise.** For each word in the input text, we randomly apply one of three noise-adding methods: a) Typo: Replace a random character with a random lowercase letter. b) Extra: Insert a random lowercase letter at a random position. c) Missing: Delete a random character. We gradually increase the noise level $\eta$. For instance, $\eta = 0.9$ means each word has a 90% probability of modification, indicating higher text corruption. Experimental results are as shown in Table 9.

We find that as $\eta$ increases, both ADCE and accuracy decrease, while AICE increases. It possible that noise likely disrupts deep structural information, forcing the model to depend on more accessible, surface-level information. This shift results in lower ADCE and higher AICE.

**Label Noise.** For the 2-digit Multiplication multiple-choice dataset, we randomly select an incorrect answer as the new correct answer. And the noise level $\eta = 0.9$ means 90% of sample labels are modified. Experimental results are as shown in Table 10.

We observe that ADCE and AICE are more robust to label noise than accuracy, showing no significant changes as noise increases. Possible reasons are (1) ADCE and AICE evaluations are based on correctly answered questions, potentially filtering out mislabeled samples before intervention. (2) Crucially, ADCE and AICE measure relative changes in model outputs pre- and post-intervention, not label accuracy as stated in Equation 5. Thus, they effectively reflect LLMs' reliance on deep or surface structures, even with label noise, provided the model shows consistent relative differences pre- and post-intervention.

