# OpenReview forum: "Beyond Surface Structure: A Causal Assessment of LLMs' Comprehension ability"
_ICLR.cc/2025/Conference — ICLR 2025 Poster_

### Official Review · Reviewer_iYaZ · 2024-11-02

**Soundness:** 3
**Presentation:** 3
**Contribution:** 3
**Rating:** 5
**Confidence:** 2

**Summary:**

This paper proposes to use the casual mediation analysis method to examine the following question: does the LLM really understand the deep semantic meaning for problem-solving or merely rely on text surface forms? Authors claim that previous methods usually investigate this effect in specific tasks without generalization. This paper presents approximated direct causal effect (ADCE) and approximated indirect causal effect (AICE) to empirically quantify the LLM dependencies on deep structure understanding or shallow surfaces.

The reviewer is not familiar with related works in this field, and therefore, the commentary below is from a general NLP researcher's point of view. The confidence score is set to 2 to reflect this point.

**Strengths:**

1. The paper presents a novel point of view to evaluate if the LM relies on the deep structures or surface forms to answer questions. Empirical and theoretical results show that the proposed ADCE is a better metric when evaluating LLM deep structure dependency.

2. The findings that closed-source LLMs rely more on deep structure while open-source models are sensitive to surface indicate that the current open-source SFT / alignment stages still need further investigation to have a higher reliance for the models on deep structure.

3. The proposed ADCE metric indicates that the accuracy of specific tasks could be misleading.

**Weaknesses:**

1. I am concerned about the necessity of applying causal mediation analysis to this research question. Mask, Replace, Swapping, etc., are common practices for augmenting text data as well as examining the sensitivity of LLMs to slight variations in the input text. The motivation for the CMA application in this submission is not clear to me, given the current writing.

2. Following 1, Section 3.3 is intuitive by itself without the theories in Section 3.1 and 3.2. If I understand correctly, Equation 5 is a simple combination of several indicator functions that reflects if intervention to the question has actual effects.

**Questions:**

Refer to the Weaknesses Part.

---

> ### Author Response · Authors · 2024-11-20
> **Rebuttal by Authors**
>
> We thank the reviewer's efforts on reviewing this paper. We now address the questions raised as follows.
>
> ---
>
> >Q1: I am concerned about the necessity of applying causal mediation analysis to this research question. Mask, Replace, Swapping, etc., are common practices for augmenting text data as well as examining the sensitivity of LLMs to slight variations in the input text. The motivation for the CMA application in this submission is not clear to me, given the current writing.
>
> Thank you for your question. We would like to clarify that (1) applying causal mediation analysis (CMA) is important to discover the LLM's deep structure understanding ability (rather than the surface one); and (2) the mask and rephrase operations in our works are specifically designed to adapt to CMA.
>
> It is true that masking, replacement, and swapping are common in LLM studies, but how to make use of these operations to reveal the LLM's deep structure understanding ability is challenging and non-trivial. Many previous research [1,2,3,4,5], performed sensitive analysis based on these operations, can only explain the surface structure understanding ability of LLM.
> In particular, these studies lacked comprehensive deep-surface structure comparisons, leading to less rigorous conclusions (see more detailed discussion in Section 1). **Simple use of operations like masking cannot isolate deep structure understanding, as it highly co-exist with the surface one.** For instance, masking ``What is 50 times 20?``  to ``What is <Mask>times 20?`` changes both deep (semantics) and surface (format) structures. **This limitation in previous studies prompt us to develop ADCE, a rigorous, CMA-based metric for assessing LLMs' deep structure comprehension.**
>
> Specifically, we rigorously formulate the relationships between input, deep structure, surface structure, and output, as a causal graph with mediation, and define the deep structure sensitivity as the direct causal effect (DCE). Then, we develop ADCE, an interpretable and computable version of DCE, using CMA and (improved) common operations like Mask. In particular, ADCE is an approximated version of DCE, and the detailed intervention operations, like masking and rephrasing, need to be specifically designed to reduce the approximation error.  For example, when estimating AICE, we mask $k$ nearby non-core words in TE instead of randomly masking. **Notably, these new designs on the intervention operations are different from the previous works.**
>
> Furthermore, we also **demonstrate the effectiveness of introducing CMA and improved common operations for quantifying the model's deep structure understanding ability through two additional experiments.** (1) We first conduct a synthetic experiment where the model's true deep structure understanding (ground truth) can be directly calculated. Within the CMA framework, we estimate the approximated model's deep structure understanding (ADCE). We find that both the trends and values of ADCE align closely with the ground truth, validating our method. Detailed results are included in the updated manuscript (``see Lines 1079-1125, Appendix E``).(2) Moreover, the spurious correlation experiments in Section 4.5 demonstrate that as spurious correlations intensify, the model's reliance on deep structures weakened. Figures 8(a) and (b) show ADCE decreasing as expected, further validating that our proposed ADCE metric, within the CMA framework, effectively quantifies LLMs' deep structure understanding ability.

---

> > ### Author Response · Authors · 2024-11-20
> > **Rebuttal by Authors**
> >
> > >Q2: Following 1, Section 3.3 is intuitive by itself without the theories in Section 3.1 and 3.2. If I understand correctly, Equation 5 is a simple combination of several indicator functions that reflects if intervention to the question has actual effects.
> >
> > Sorry for the confusion caused. **Equation (5) is not only unintuitive, but it is also derived through a step-by-step analysis in Sections 3.1, 3.2, and 3.3** , resulting in a computable and interpretable version of the well-known direct causal effect (DCE). Specifically:
> > - First, after establishing in Section 1 that LLMs' understanding of deep structures cannot be directly quantified (as pointed out in the response to Q1), **Section 3.1** defines LLMs' deep structure understanding ability as DCE, based on a given causal graph with mediation.
> > - Then, based on the given causal graph in Section 3.1, we propose **Equation (3)** to evaluate DCE, which relies on the classic CMA framework.
> > - Next, to address unobservable data in classic CMA frameworks, **Section 3.2** proposes ADCE, an approximated version of DCE in **Equation (4)**. To minimize approximation losses, **Section 3.3** introduces careful intervention strategies to ensure accuracy. These include masking $k$ nearby neighbor non-core words in TE and using special prompts, such as ``modify the keywords with minimal word changes``, for minimal word changes when transforming TE to AICE.
> > - Finally, **Section 3.2** further addresses computational problems arising from LLMs' outputs not always being in numerical form. Referring to previous research [1,7], we introduce indicator functions in **Equation (4)**, ultimately arriving at the computable and interpretable **Equation (5)**.
> >
> > Equation (5) relies on rigorous analysis in Sections 3.1, 3.2 and 3.3, which provides a computable and interpretable metric (ADCE) as an effective approximation of LLMs' deep structure understanding ability. It is neither simple nor intuitive.
> >
> > Thank you for your feedback again. In the updated manuscript, we have accordingly adjusted Section 3.2 to ensure a tighter structure among Sections 3.1, 3.2, and 3.3, more clearly demonstrating the derivation of Equation (5) (``see Line 266-278``).
> >
> > [1] Stolfo, Alessandro, et al. "A Causal Framework to Quantify the Robustness of Mathematical Reasoning with Language Models." Proceedings of the 61st Annual Meeting of the Association for Computational Linguistics, Volume 1: Long Papers. Association for Computational Linguistics, 2023.
> >
> > [2] Ashish Hooda, Mihai Christodorescu, Miltos Allamanis, Aaron Wilson, Kassem Fawaz, and Somesh Jha. Do large code models understand programming concepts? a black-box approach. arXiv preprint arXiv:2402.05980, 2024.
> >
> > [3] Gonzalez, Javier, and Aditya V. Nori. "Does Reasoning Emerge? Examining the Probabilities of Causation in Large Language Models." The Thirty-eighth Annual Conference on Neural Information Processing Systems.
> >
> > [4] Siyuan Guo, Aniket Didolkar, Nan Rosemary Ke, Anirudh Goyal, Ferenc Huszar, and Bernhard Scholkopf. Learning beyond pattern matching? assaying mathematical understanding in llms. arXiv preprint arXiv:2405.15485, 2024.
> >
> > [5] Bowen Jiang, Yangxinyu Xie, Zhuoqun Hao, Xiaomeng Wang, Tanwi Mallick, Weijie J Su, Camillo Jose Taylor, and Dan Roth. 2024. A Peek into Token Bias: Large Language Models Are Not Yet Genuine Reasoners. In Proceedings of the 2024 Conference on Empirical Methods in Natural Language Processing, pages 4722–4756, Miami, Florida, USA. Association for Computational Linguistics.
> >
> > [6] Meiqi Chen, Yixin Cao, Yan Zhang, and Chaochao Lu. 2024. Quantifying and Mitigating Unimodal Biases in Multimodal Large Language Models: A Causal Perspective. In Findings of the Association for Computational Linguistics: EMNLP 2024, pages 16449–16469, Miami, Florida, USA. Association for Computational Linguistics.
> >
> > ---
> > We hope the above response can resolve your concern. If you find our response helpful, please consider raising the score for support of our work. We are open to discuss more if any question still hold.

---

> > > ### Comment · Reviewer_iYaZ · 2024-11-20
> > >
> > > I appreciate the reviewer's feedback on the concerns proposed regarding the necessity of causal mediation analysis. After reading the revised version of this submission, I updated my ratings accordingly.

---

> ### Author Response · Authors · 2024-11-21
> **Response by Authors**
>
> Dear Reviewer iYaZ,
>
> Thank you for acknowledging our rebuttal. We are delighted to see that our response has addressed your previous concerns regarding CMA. Given your current score, we would greatly appreciate if you could share any further concerns you may have. Please share them with us, and we are willing to address and discuss these issues in more detail to improve our work.
>
> Authors

---

### Official Review · Reviewer_zxFT · 2024-11-04

**Soundness:** 4
**Presentation:** 3
**Contribution:** 4
**Rating:** 6
**Confidence:** 3

**Summary:**

This paper presents a causal mediation analysis framework to assess LLMs' comprehension ability of deep structure versus surface structure. The authors propose ADCE and AICE metrics to quantify comprehension of deep and surface structures. They demonstrate that most LLMs exhibit genuine deep structure comprehension that increases with model scale, while dependence on surface structure varies between open and closed-source models.

**Strengths:**

- Paper is well-written.
- Comprehensive empirical evaluation across multiple tasks and model families with insightful findings about deep vs surface structure reliance
- This paper is novel; proposes ADCE and AICE for quantification based on causal mediation analysis.
- The method is task-agnostic; evaluation on 5 tasks across math, logic, common sense benchmarks.

**Weaknesses:**

- The relationship between ADCE and fine-tuning (Section 4.3) is only briefly explored.

**Questions:**

- ADCE seems to capture both sufficiency and necessity of deep structure changes in causing output variations. How might different post training strategies influence a model's reliance on deep vs. surface structure?

---

> ### Author Response · Authors · 2024-11-20
> **Rebuttal by Authors**
>
> Thanks for the reviewer's constructional feedback on this paper. We now address the questions as follows.
>
> ---
>
> > Q1: The relationship between ADCE and fine-tuning (Section 4.3) is only briefly explored.
>
> Thank you for your question. Section 4.3 investigates why ADCE shows anomalous values (smaller than 0) for certain tasks and LLMs, such as Llama-3-8b on Analytic Entailment. We hypothesize this is due to unactivated or absent relevant knowledge in the pre-training data, rather than a failure of ADCE. To test this, we conduct supervised fine-tuning (SFT). The results show a significant increase in ADCE after SFT, supporting our hypothesis that the anomalous pre-SFT ADCE values reflect a lack or inactivation of task-specific knowledge in the LLMs, not a deficiency in ADCE.
>
> We acknowledge that exploring SFT from an ADCE perspective is an intriguing topic, but it is beyond the scope of this paper and we leave it for future exploration. And if you have specific suggestions for analyzing SFT via ADCE, we are very willing to explore this topic further.
>
> > Q2: ADCE seems to capture both sufficiency and necessity of deep structure changes in causing output variations. How might different post training strategies influence a model's reliance on deep vs. surface structure?
>
> Thanks for mentioning this interesting question. We have expanded our analysis to include two additional post-training approaches: Instruction Fine-Tuning (IFT) [1] and Fine-Tuning with In-Context Learning (FTICL) [2]. We've also analysed the In-Context Learning (ICL) [3] method, due to its effectiveness in harnessing the models' inherent abilities to comprehend and produce responses, as well as its popularity within the NLP community.
>
> While other post training strategie like DPO [4] and RLHF [5] are often mentioned in the training of LLM for aligning with human preferences, these methods require constructing paired training data. Given that preference alignment is not our main focus and considering the time constraints for additional dataset generation, we leave the exploration of these methods to future studies.
>
> Following the experimental setting in Section 4.5, we consider Llama-3-8b on the Analytic Entailment task. We add more experimental details in`` Appendix G.2, Lines 1330 - 1387`` of our updated manuscript and summarize the main results in the following table.
>
> | Metric | Pre-training | SFT |IFT |FTICL |ICL |
> | -------- | -------- | -------- |-------- |-------- |-------- |
> | Accuracy    | 0.457     | 0.743     |  0.800   | 0.786   |0.771     |
> | ADCE    |  -0.071    |   0.318  |   0.478 |   0.533 |   0.455  |
>
> We find that various post-training strategies and ICL all lead to improvements in both model accuracy and deep structure understanding ability (ADCE). Moreover, FTICL and IFT, which consider both prompt engineering and parameter optimization, yield greater gains compared to SFT, which only focuses on parameter optimization, or ICL, which only optimizes prompts.
>
>
> [1] Wei, Jason, et al. "Finetuned Language Models are Zero-Shot Learners." International Conference on Learning Representations.
>
> [2] Anil C, Wu Y, Andreassen A, et al. Exploring length generalization in large language models[J]. Advances in Neural Information Processing Systems, 2022, 35: 38546-38556.
>
> [3] Brown T B. Language models are few-shot learners[J]. arXiv preprint arXiv:2005.14165, 2020.
>
> [4] Rafailov R, Sharma A, Mitchell E, et al. Direct preference optimization: Your language model is secretly a reward model[J]. Advances in Neural Information Processing Systems, 2024, 36.
>
> [5] Ouyang L, Wu J, Jiang X, et al. Training language models to follow instructions with human feedback[J]. Advances in neural information processing systems, 2022, 35: 27730-27744.
>
> ----
>
> We hope our response and revisions have addressed your concerns. If you find our revisions and response helpful, we would greatly appreciate your consideration in raising the score to better support our work. We remain open to further discussion if you have any remaining questions.

---

> > ### Author Response · Authors · 2024-11-25
> >
> > Dear Reviewer zxFT,
> >
> > We extend our thanks for your time and advice, as they have helped make our paper more comprehensive. In response to your questions about ADCE's performance on various post-training strategies, we have provided additional experiments which have been incorporated into the latest manuscript. We genuinely hope that these responses effectively address your concerns. If you have any additional questions or suggestions, please feel free to let us know. We are more than willing to engage in further discussions with you regarding this work.
> >
> > Best,
> >
> > Authors

---

### Official Review · Reviewer_jhyn · 2024-11-04

**Soundness:** 3
**Presentation:** 3
**Contribution:** 3
**Rating:** 6
**Confidence:** 3

**Summary:**

The paper introduces a framework to evaluate LLMs by distinguishing between their reliance on deep structures and surface structures. Using causal mediation analysis, it proposes metrics called Approximated Direct Causal Effect for deep structure comprehension and Approximated Indirect Causal Effect for surface structure, offering a more nuanced assessment of model understanding than accuracy alone. The findings reveal that closed-source models, like GPT, rely more on deep structure, whereas open-source models, like Llama, are more sensitive to surface structures, though this sensitivity decreases with model size

**Strengths:**

1. The paper introduces a unique framework that goes beyond accuracy to assess how language models rely on deep versus surface structures.
2. The ADCE and AICE metrics are interpretable and allow for precise distinctions between models' reliance on core semantics and surface-level structures.
3. The framework highlights significant differences between closed-source and open-source models in their reliance on deep vs. surface structures, contributing valuable insights into model development trends.

**Weaknesses:**

1. Although the metrics provide valuable insights, their exact interpretations may vary depending on model architecture, making it difficult to generalize findings across diverse LLMs without further context.
2. The approach may require internal access to models for accurate analysis, potentially limiting its use with proprietary or black-box models where such transparency isn’t feasible.

**Questions:**

1. Could the authors clarify how practitioners might interpret ADCE and AICE values across different model architectures?
2. How does the approach perform on NLP tasks involving noisy or unstructured data?

---

> ### Author Response · Authors · 2024-11-20
> **Rebuttal by Authors**
>
> We thank the reviewer's acknowledgment on this paper. We now address the questions raised as follows.
>
> ---
>
> > Q1: Although the metrics provide valuable insights, their exact interpretations may vary depending on model architecture... / Could the authors clarify how practitioners might interpret ADCE and AICE values across different model architectures?
>
> Thanks for mentioning this important question. The proposed metrics ADCE and AICE are model-agnostic, treating models as black boxes and focusing only on their outputs.  We then demonstrate why the proposed metrics are model-agnostic from both experimental and methodological perspectives.
>
> 1. Experimentally, we conduct additional experiments to **demonstrate the consistent properties of these metrics across different model architectures.** Taking ADCE as an example, we extend the experiments in Section 4.5 to include different model architectures. Due to computational constraints, we primarily focus on Llama-3 (Transformer) and Mistral-7b (Transformer with Sliding Window Attention). We observe that as spurious correlations increased, all ADCEs show consistent decreasing trends (↓) across different LLM architectures while Accuracy shows almost no significant change (-), indicating ADCE's architecture-independent nature.
>
> | Model   Architecture  | Metric   | $n_{majority}$ (%) ||||
> |:------------|:---------|:---:|:---:|:---:|:---:|
> |             |      Expected True causal effect of $d$ (&darr;)    | 50 | 70 | 90 | 100 |
> | Llama-3-8b  | ADCE (&darr;)    | 0.890| 0.866| 0.773| 0.348|
> |             | Accuracy (-) | 0.961| 0.944| 0.902| 0.974|
> | Llama-3-70b | ADCE   (&darr;)     | 0.840| 0.877| 0.697| 0.599|
> |             | Accuracy  (-)| 0.957| 0.935| 0.931| 0.952|
> | Mistral-7b  | ADCE  (&darr;)      | 0.701| 0.727| 0.552| 0.545|
> |             | Accuracy  (-)| 0.939| 0.913| 0.947| 0.957|
>
> 2. Methodologically, **the calculations of ADCE and AICE are independent of model architecture**. As shown in Equation (5) (``Line 283 in Section 3.2``), we only need the model outputs before intervention $Y^{origin}$ and after interventions $Y(T=1, s(T=1))$ and $Y(T=0, s(T=0))$ to calculate these metrics, without relying on specific architectural details.
>
> > Q2: The approach may require internal access to models for accurate analysis, potentially limiting its use with proprietary or black-box models where such transparency isn’t feasible.
>
> Thank you for your question. Our method does not require internal access to models for accurate analysis. The interventions in our method are performed exclusively on the inputs, without any internal access to the models. As for metric calculation, our method focuses solely on the outputs of LLMs, making our approach applicable to both black-box and white-box models. Specifically, as indicated in Equation (5) of Section 3, ADCE and AICE only require comparing LLMs' outputs in three scenarios: before intervention ($Y^{origin}$), after intervention on both deep and surface structures ($Y(T=1, s(T=1))$), and after intervention only on surface structure ($Y(T=0, s(T=0))$). This comparison is made without the need for any additional internal information about the models' architectures.
>
> > Q3: How does the approach perform on NLP tasks involving noisy or unstructured data?
>
> Thanks for this great suggestion.
> - **For unstructured data**, the datasets used in the experiments, such as CommonsenseQA, Analytic Entailment, and GSM8k, do not have a uniform text template and are not structured data. The experimental results in Section 4 have demonstrated the effectiveness of ADCE and AICE on these datasets. More details about the datasets can be found in Appendix A (``Lines 758-787``).
> - **For noisy data**, we consider two scenarios: text noise [1, 2, 3] and label noise [4, 5]. Using the 2-digit Multiplication dataset and LLama-3-8b as an example:
>
> (1) Text Noise: For each word in the input text, we randomly apply one of three noise-adding methods: a) Typo: Replace a random character with a random lowercase letter. b) Extra: Insert a random lowercase letter at a random position. c) Missing: Delete a random character. We gradually increase the noise level $\eta$. For instance, $\eta=0.9$ means each word has a 90% probability of modification, indicating higher text corruption. The experimental results are as follows:
>
> | $\eta$ | Accuracy| ADCE|AICE|
> | -------- | -------- | -------- |-------- |
> | 0    | 0.710     | 0.733     |0.264     |
> | 0.2    | 0.497    | 0.681     |0.319    |
> | 0.5    |   0.201  | 0.550 |  0.448  |
> | 0.7    |   0.093  |   0.438  |   0.556  |
> | 0.9    |  0.031  |    0.444 |   0.556  |
>
> As the noise level $\eta$ increases, both ADCE and Accuracy decrease, while AICE increases. It possible that noise likely disrupts deep structural information, forcing the model to depend on more accessible, surface-level information. This shift results in lower ADCE and higher AICE.

---

> > ### Author Response · Authors · 2024-11-20
> > **Rebuttal by Authors**
> >
> > (2) Label Noise: For the 2-digit Multiplication multiple-choice dataset, we randomly select an incorrect answer as the new correct answer. And the noise level $\eta=0.9$ means 90% of sample labels are modified. Experimental results are as follows:
> >
> > | $\eta$| Accuracy| ADCE|AICE|
> > | -------- | -------- | -------- |-------- |
> > | 0    | 0.710     | 0.733     |0.264     |
> > | 0.2    |   0.589 |   0.787  |  0.213 |
> > | 0.5    |   0.352  | 0.793 |  0.207  |
> > | 0.7    |   0.224  |  0.757   |  0.241 |
> > | 0.9    | 0.070   |  0.771   |  0.229   |
> >
> > We observe that ADCE and AICE are more robust to label noise than accuracy, showing no significant changes as noise increases. Possible reasons are (1) ADCE and AICE evaluations are based on correctly answered questions, potentially filtering out mislabeled samples before intervention. (2) Crucially, ADCE and AICE measure relative changes in model outputs before and after intervention, rather than label accuracy as stated in Equation (5). Thus, they effectively reflect LLMs' reliance on deep or surface structures, even with label noise, provided the model shows consistent relative differences before and after intervention.
> >
> > We appreciate your suggestion again. We have added the experiments on noisy data to ``Appendix I , Lines 1458 - 1508`` in the updated manuscript.
> >
> > [1] Karpukhin V, Levy O, Eisenstein J, et al. Training on synthetic noise improves robustness to natural noise in machine translation[C] Proceedings of the 5th Workshop on Noisy User-generated Text (W-NUT 2019). 2019: 42-47.
> >
> > [2] Belinkov, Yonatan, and Yonatan Bisk. "Synthetic and Natural Noise Both Break Neural Machine Translation." International Conference on Learning Representations. 2018.
> >
> > [3] Wei, Jason, and Kai Zou. "EDA: Easy Data Augmentation Techniques for Boosting Performance on Text Classification Tasks." Proceedings of the 2019 Conference on Empirical Methods in Natural Language Processing and the 9th International Joint Conference on Natural Language Processing (EMNLP-IJCNLP). 2019.
> >
> > [4] Wu, Tingting, et al. "NoisywikiHow: A Benchmark for Learning with Real-world Noisy Labels in Natural Language Processing." Findings of the Association for Computational Linguistics: ACL 2023. 2023.
> >
> > [5] Garg S, Ramakrishnan G, Thumbe V. Towards robustness to label noise in text classification via noise modeling[C]//Proceedings of the 30th ACM International Conference on Information & Knowledge Management. 2021: 3024-3028.
> >
> > ---
> > We hope the above response can address your concern. If you find our revise and response helpful, please consider raising the score for better support of our work. We are open to discuss more if any question still hold.

---

> > > ### Author Response · Authors · 2024-11-25
> > >
> > > Dear Reviewer jhyn,
> > >
> > > We would like to express our appreciation for your feedback and the time you have dedicated, as it has greatly contributed to improving the clarity and quality of our manuscript. Based on your suggestions, we have expanded our experiments to include more scenarios, including additional model architectures and noisy data. We have also provided more explanations on why ADCE is considered a black-box method. All these updates have been added to the latest manuscript. We sincerely hope that our response has effectively addressed your concerns. Should you have any additional questions or suggestions, please do not hesitate to inform us. We eagerly anticipate engaging in further discussions with you on this work.
> > >
> > > Best,
> > >
> > > Authors

---

### Official Review · Reviewer_WdDi · 2024-11-06

**Soundness:** 3
**Presentation:** 3
**Contribution:** 3
**Rating:** 6
**Confidence:** 3

**Summary:**

The manuscript investigates the comprehension abilities of large language models (LLMs) beyond superficial structures by introducing a causal mediation framework. The authors propose the use of approximated direct causal effect (ADCE) and approximated indirect causal effect (AICE) as proxies to quantify comprehension of deep and surface structures, respectively. Through this framework, they empirically evaluate the reliance of mainstream LLMs, such as GPT and Llama, on deep versus surface structures across a range of tasks, including mathematical reasoning and commonsense understanding.

**Strengths:**

+ The paper presents an innovative method to quantify the comprehension of deep structures in LLMs through causal analysis.
+ Comprehensive datasets and models are used for evaluation.
+ Experimental results across multiple tasks suggest that ADCE is a valuable metric for assessing model comprehension.

**Weaknesses:**

- It is unclear how deep structures are accurately identified and separated from surface structures, which is crucial for the validity of the interventions.
- The ADCE and AICE are presented as approximations, but it is unclear how accurately they reflect the true causal effects intended for evaluation.

**Questions:**

Overall, I appreciate the authors’ efforts in introducing a novel approach to assessing the comprehension ability of LLMs, especially in quantifying deep structure reliance. However, I have a few questions and points for clarification:

- Your approach relies on interventions targeting deep and surface structures. In your experiments, how are deep structures accurately identified and separated from surface structures? It would be helpful to understand the criteria or methods used to make this distinction reliably.

- Since ADCE and AICE are approximations, have you conducted any experiments to validate these metrics against true causal effects? Any validation results would help clarify the accuracy and reliability of these metrics in capturing the LLMs’ reliance on different structural components.

- I have a question regarding the use of causal mediation analysis within the causal structure presented in Figure 3.
As I understand it, causal mediation analysis is typically used to assess how the effect of a treatment variable X on an outcome Y is mediated through an in intermediate variable Z. This approach is often applied to measure both the direct effect of X on Y and the indirect effect mediated by Z (i.e., X -> Z -> Y). For example, a drug X might have a direct effect on a disease Y, but it may also cause patients to take aspirin Z, which further impacts Y through this indirect pathway.
In your causal graph, however, it seems that the deep structure d and surface structure s within the input x, independently affect the outcome Y through separate pathways (d -> Y and s -> Y). Since d does not mediate or influence s, these pathways appear to act in parallel rather than in a sequential manner where one variable mediates the effect of another.
Given this structure, could you please clarify whether causal mediation analysis is appropriate in this context? It would be helpful to understand how the direct and indirect effects are conceptualized in this framework, especially if the mediation structure does not strictly follow the conventional X -> Z -> Y pathway.

---

> ### Author Response · Authors · 2024-11-20
> **Rebuttal by Authors**
>
> Thanks for your time reviewing this paper. We now address your questions as follows.
>
> ---
> > Q1: It is unclear how deep structures are accurately identified and separated from surface structures. / Your approach relies on interventions targeting deep and surface structures. In your experiments, how are deep structures accurately identified and separated from surface structures?
>
> This is a good question. Strictly differentiating the impact of deep and surface structures on output in experiments is not easy, as changes in deep structure often inevitably lead to changes in surface structure. For instance, consider the example in Table 1: when we mask ``50`` in the question``What is 50 times 20`` to create ``What is <Mask> times 20?``, we alter both the deep and surface structures simultaneously.
>
> The high coupling between deep and surface structures makes distinguishing them experimentally hard. Therefore, **our key idea is to design ADCE and AICE as indicators highly correlated with LLMs' deep and surface structure understanding capabilities, respectively**. Specifically, to ensure that AICE is solely related to surface structure, AICE is proposed to compare changes in LLMs' outputs when only the surface structure is altered. On the other hand, ADCE uses an indirect method, approximating deep structure impact by subtracting surface structure impact (AICE) from the total effect (TE) of both structures in Equation (5). To minimize approximation errors and ensure that ADCE is solely highly correlated with deep structure, we carefully design intervention strategies in Section 3.3. For instance, based on the intervention inputs in TE, we further mask the $k$ non-core semantic words closest to the masked core semantic word in TE as the intervention text for AICE (see more details in ``Lines 306-309, Section 3.3``). These strategies ensure ADCE primarily measures output changes caused by alterations in deep structure, effectively capturing the model's deep understanding ability.
>
> Experimentally, both synthetic and spurious correlation studies (detailed below in Points 1 and 2, Question 2) confirm that ADCE and AICE strongly correlate with the model's true understanding of deep and surface structures, respectively. This validates our metrics' ability to distinguish between the impacts of deep and surface structures on model outputs.
>
> > Q2: The ADCE and AICE are presented as approximations, but it is unclear how accurately they reflect the true causal effects... / Since ADCE and AICE are approximations, ... validate these metrics against true causal effects?
>
> Thanks for mentioning this important question. We conduct the following two experiments to demonstrate the validity of ADCE and AICE:
>
> 1. Experiments on synthetic data with known true causal effects.
>
> We reduce the causal relationships between deep structure $d$, surface structure $s$, input $x$, and output $y$ shown in Figure 3 to a simplified Structural Causal Model (SCM) [1]. In this simplified model, $d$, $s$ and $x$ are all scalars with linear relationships. Despite simplification, this SCM retains the key causal relationships in Figure 3, where $x$'s effect on  is mediated through two parallel paths: $x \to d \to y$ and $x \to s \to y$. This simplification allows training a logistic regression with explicit functions, enabling direct computation of true causal effects for $d$ and $s$, thus facilitating ADCE and AICE validation. The SCM is defined as:
>
> $$
> \\begin{aligned}
> x &\\sim \\mathcal{N}(0, 1),\\quad d = x + \\epsilon_d, \\quad s = x + \\epsilon_s.\\\\
> y &= \\begin{cases}
> 1, & \\text{if } \\sigma(c_1\\cdot d + c_2 \\cdot s + \\epsilon_y) > 0.5 \\\\
> 0, & \\text{otherwise}
> \\end{cases}
> \\end{aligned}
> $$
> where independent small noises $\epsilon_d \sim \mathcal{N}(0, 0.25)$, $\epsilon_s \sim \mathcal{N}(0, 0.25)$ and $\epsilon_y \sim \mathcal{N}(0, 1)$. $\sigma(\cdot)$ is Sigmoid function. $c_1$ and $c_2$ are weight parameters for $d$ and $s$, respectively.
>
> In the updated manuscript, we provide more details about synthetic experiments and visualize how true causal effects, ADCE, and AICE trends change with the weight parameters $c_1$  (``see Lines 1079-1125, Appendix E``).
>
> Here, **we summarize main messages** :
> - The trends in Figure 11 (a) of approximated causal effects (ADCE and AICE) and true causal effects of $d$ and $s$ on $y$ align as $c_1$ increases, demonstrating the efficacy of ADCE and AICE.
> - The trend in Figure 11 (a) shows the difference between approximated causal effects (ADCE - AICE) aligns perfectly with the trend of the difference between true causal effects (True CE of $d$ - True CE of $s$). This consistency validates our key experiment in Section 4.4, comparing LLMs' understanding of deep versus surface structures.
> - We normalize approximated and true causal effects to $[0,1]$, eliminating numerical incomparability. This reveals a much closer alignment between their trends, as shown by the nearly identical curves in Figure 11(b).

---

> > ### Author Response · Authors · 2024-11-20
> > **Rebuttal by Authors**
> >
> > 2. Experiments on LLMs with known true causal effect trend.
> >
> > In scenarios where true causal effects in LLMs are unknowable, we focus on expected trends to validate proposed metrics. Extending spurious correlation experiments in Section 4.5 to more LLMs (e.g., Llama-3 and Mistral), we expect ADCE to decrease (&darr;) as spurious correlations level ( $n_{majority}$ (%)) increases, indicating models' decreased reliance on deep structures due to spurious correlation. The table below shows ADCE decreasing trends(&darr;) across diverse LLMs aligning with our expectation while the metric Accuracy shows almost no significant change (-), indirectly supporting our metrics' validity.
> >
> > | LLMs  | Metric   | $n_{majority}$ (%) ||||
> > |:------------|:---------|:---:|:---:|:---:|:---:|
> > |             |   Expected True causal effect of $d$  (&darr;)  | 50 | 70 | 90 | 100 |
> > | Llama-3-8b  | ADCE (&darr;)    | 0.890| 0.866| 0.773| 0.348|
> > |             | Accuracy (-) | 0.961| 0.944| 0.902| 0.974|
> > | Llama-3-70b | ADCE   (&darr;)     | 0.840| 0.877| 0.697| 0.599|
> > |             | Accuracy  (-)| 0.957| 0.935| 0.931| 0.952|
> > | Mistral-7b  | ADCE  (&darr;)      | 0.701| 0.727| 0.552| 0.545|
> > |             | Accuracy  (-)| 0.939| 0.913| 0.947| 0.957|
> >
> > > Q3: I have a question regarding the use of causal mediation analysis within the causal structure presented in Figure 3... This approach is often applied to measure both the direct effect of X on Y and the indirect effect mediated by Z (i.e., X -> Z -> Y)... however, it seems that the deep structure d and surface structure s within the input x, independently affect the outcome Y through separate pathways (d -> Y and s -> Y)... Given this structure, could you please clarify whether causal mediation analysis is appropriate in this context?.. how the direct and indirect effects are conceptualized in this framework, especially if the mediation structure does not strictly follow the conventional X -> Z -> Y pathway.
> >
> > Thanks for pointing out the difference. The causal structure in this paper indeed differs from the traditional mediation model ($x \to z \to y$ and $x \to y$). Instead, we employ a variant of the classic causal mediation model known as the Parallel Multiple Mediator Model [2,3,4]. In our model, the deep structure ($d$) and surface structure ($s$) serve as two parallel mediators for the input x. The specific causal paths can be represented as $\mathbf{x} \to d \to Y$ and $\mathbf{x} \to s \to Y$.
> >
> >
> > Despite structural differences, our parallel multiple mediator model aligns with traditional mediation models in key aspects. Like classic mediation models, we also can decompose the total causal effect (TE: $\mathbf{x}  \to Y$) into two parallel pathways and define: a direct causal effect (DCE: $\mathbf{x}  \to d \to Y$) through our variable of interest (deep structure $d$), and an indirect causal effect (ICE: $\mathbf{x} \to s \to Y$) through the mediator (surface structure $s$). This decomposition mirrors the parallel $x \to y$ and $x \to z \to y$ paths in traditional models and ensures that the relationship between TE, ICE, and DCE in Equation (3) holds. Additionally, our model satisfies key assumptions of causal mediation analysis as discussed in Appendix B.1. This fundamental consistency enables the application of established causal mediation methods to our model.
> >
> > We appreciate the reviewer's question again. And we have added above discussion in the revised manuscript to further clarify why causal mediation analysis is applicable to our framework (``Lines 809-832, Appendix B``).
> >
> > [1] Judea Pearl. Causality. Cambridge university press, 2009
> >
> > [2] Preacher K J, Hayes A F. Asymptotic and resampling strategies for assessing and comparing indirect effects in multiple mediator models[J]. Behavior research methods, 2008, 40(3): 879-891.
> >
> > [3] Bolin J H. Introduction to mediation, moderation, and conditional process analysis: a regression-based approach[J]. 2014.
> >
> > [4] VanderWeele T, Vansteelandt S. Mediation analysis with multiple mediators[J]. Epidemiologic methods, 2014, 2(1): 95-115.
> >
> > ---
> >
> > We hope the above response can resolve your concern. If you find our revise and response helpful, please consider raising the score to support our work. Also please let us know if there is any more question.

---

> > > ### Author Response · Authors · 2024-11-25
> > >
> > > Dear Reviewer WdDi,
> > >
> > > We extend our thanks for your time and feedback, as they have contributed to improving the quality of our paper. In light of your concerns about the accuracy of ADCE, we have conducted additional experiments and provided explanations on the separation of deep and surface structure and the causal mediation analysis. These updates have been incorporated into the manuscript accordingly. We genuinely hope that these responses effectively address your concerns. If you have any further questions or issues, please do not hesitate to reach out to us. We eagerly look forward to the opportunity to continue the discussion on this work.
> > >
> > > Best,
> > >
> > > Authors

---

> > > > ### Comment · Reviewer_WdDi · 2024-12-01
> > > > **Thanks for the rebuttal**
> > > >
> > > > I appreciate the reviewer's feedback concerning the additional experiments and the clarifications provided on the separation of deep and surface structures, as well as on the causal mediation analysis. After reviewing the updated submission, I have adjusted my ratings accordingly.

---

> > > > > ### Author Response · Authors · 2024-12-01
> > > > >
> > > > > Dear Reviewer WdDi,
> > > > >
> > > > > Thank you for your feedback. We're glad our response addressed your concerns, and we appreciate the updated score.
> > > > >
> > > > > Best,
> > > > >
> > > > > Authors

---

### Author Response · Authors · 2024-12-03

Dear Area Chair and Reviewers,

As the discussion period comes to an end, we sincerely thank all for your time and effort in examining and progressing our work. Your constructive feedback has enhanced the quality of our research.

It's encouraging to see that all reviewers  commended the novelty of our work  (Reviewers WdDi, jhyn, zxFT, iYaZ) . Our proposed metrics, ADCE and AICE, are recognized as interpretable (Reviewer jhyn), task-agnostic (Reviewer zxFT), superior to accuracy in specific tasks (Reviewer iYaZ), and are valuable for detailed assessment of LLMs' comprehension abilities (Reviewers WdDi, jhyn, zxFT, iYaZ). We also appreciate that reviewers found our experiments comprehensive (Reviewers zxFT, WdDi), offering insightful findings into LLMs (Reviewers jhyn, zxFT, iYaZ).

Furthermore, we would like to express our gratitude for the reviewers' thoughtful suggestions. Specifically, the requests for additional clarification from Reviewers WdDi and iYaZ, such as ensuring distinguishability between deep and surface structures, guaranteeing ADCE's accuracy, and justifying the necessity of causal mediation analysis, have helped enhance the clarity of our paper and the credibility of our method. Reviewers jhyn and zxFT suggested extending our approach to more diverse scenarios, including different model architectures, noisy data, and additional post-training strategies. These supplementary experimental results have validated the effectiveness of our proposed metrics, ADCE and AICE, across various tasks. We have incorporated these insightful suggestions into our latest manuscript.

We understand the work pressure faced by the Area Chair and Reviewers in a highly competitive conference like ICLR. We would like to express our gratitude once again to the Area Chair and Reviewers for their feedback and engagement with our work during both the review and discussion phases.

Best,

Authors

---

### Meta-Review · Area_Chair_F3zk · 2024-12-21

**Metareview:**

The manuscript examines the comprehension abilities of LLMs using a causal mediation analysis framework. The authors introduce well-designed ADCE and AICE metrics to quantify comprehension at both deep and surface levels. Extensive experiments across multiple tasks and foundation models, along with thorough results analysis, provide valuable insights into evaluating LLMs.

The authors did a good job in the rebuttal such as clarifying the causal structure and providing further analysis of experimental results. Overall, this work meets the acceptance criteria and I recommend its acceptance. For the next version, please revise carefully according to the reviewers' comments.

**Additional Comments On Reviewer Discussion:**

Discussion Summary:
- Further clarifications:
  - The causal structure of this work.
  - How metrics ADCE and AICE reflect the true causal effects intended for evaluation.
- Experimental Analysis:
  - NLP tasks involving noisy or unstructured data.
  - The influence of post-training strategies.

The authors did a good job in the rebuttal. Please improve the work based on the provided comments.

---

### Decision · Program_Chairs · 2025-01-22

Accept (Poster)